# Revealing solid electrolyte interphase formation through interface-sensitive *Operando* X-ray absorption spectroscopy

Jack E. N. Swallow[1,2,3], Michael W. Fraser[1,3], Nis-Julian H. Kneusels [4], Jodie F. Charlton[1,2], Christopher G. Sole[2,3], Conor M. E. Phelan[1], Erik Björklund [1,3], Peter Bencok[2], Carlos Escudero [5], Virginia Pérez-Dieste[5], Clare P. Grey [4], Rebecca J. Nicholls[1] & Robert S. Weatherup [1,2,3] ✉

The solid electrolyte interphase (SEI) that forms on Li-ion battery anodes is critical to their long-term performance, however observing SEI formation processes at the buried electrode-electrolyte interface is a significant challenge. Here we show that *operando* soft X-ray absorption spectroscopy in total electron yield mode can resolve the chemical evolution of the SEI during electrochemical formation in a Li-ion cell, with nm-scale interface sensitivity. O, F, and Si K-edge spectra, acquired as a function of potential, reveal when key reactions occur on high-capacity amorphous Si anodes cycled with and without fluoroethylene carbonate (FEC). The sequential formation of inorganic (LiF) and organic (-(C=O)O-) components is thereby revealed, and results in layering of the SEI. The addition of FEC leads to SEI formation at higher potentials which is implicated in the rapid healing of SEI defects and the improved cycling performance observed. *Operando* TEY-XAS offers new insights into the formation mechanisms of electrode-electrolyte interphases and their stability for a wide variety of electrode materials and electrolyte formulations.

Lithium-ion batteries (LIBs) are the dominant technology for powering portable electronic devices[1], and increasingly used as the power source for electric vehicles[2,3], where they can be charged using electricity supplied from renewable energy sources, making them a critical component in efforts to decarbonize transport. To encourage more widespread adoption of LIB-powered vehicles and extend their use to areas such as heavy transport, further improvements in energy density (gravimetric capacity × voltage) are needed while maintaining cycle lifetime[3–5]. This has driven the search for electrode materials with higher capacities and which expand the operating voltage window of LIBs. However, the interfacial stability of these materials when in contact with the electrolyte is a critical consideration in maintaining

cycle lifetime. Ongoing irreversible reactions otherwise lead to poor coloumbic efficiencies, loss of active lithium, and premature cell-failure. Understanding these interfacial reactions is a major challenge in the development of improved battery materials, and thus interface-sensitive *operando* characterization techniques are needed to resolve the processes occurring in working batteries.

Graphite is the most widely used anode material in LIBs, where its low lithiation/delithiation potential (~0.1 V vs. Li/Li⁺) and reasonably high gravimetric capacity (372 mAhg⁻¹) contribute to achieving high energy densities[6–9]. However, lithiation/delithiation occurs below the reduction potential of conventional electrolytes based on $LiPF_6$ dissolved in mixtures of ethylene carbonate (EC) and linear alkyl

[1]Department of Materials, University of Oxford, Parks Road, Oxford OX1 3PH, UK. [2]Diamond Light Source, Didcot, Oxfordshire OX11 0DE, UK. [3]The Faraday Institution, Quad One, Harwell Science and Innovation Campus, Didcot OX11 0RA, UK. [4]Department of Chemistry, University of Cambridge, Lensfield Road, Cambridge CB2 1EW, UK. [5]ALBA Synchrotron Light Source, Carrer de la Llum 2-26, 08290 Cerdanyola del Vallès, Barcelona, Spain. ✉e-mail: robert.weatherup@materials.ox.ac.uk

carbonates such as dimethyl carbonate (DMC). Graphite's successful application thus relies in large-part on the formation of a stable, electrically insulating, yet Li+- permeable solid-electrolyte interphase (SEI) by electrolyte decomposition[10,11]. Although SEI formation inherently involves significant irreversible reactions during the initial cycles, ideally it then serves to passivate the anode surfaces against further electrolyte decomposition allowing continuing operation outside of the electrolyte's stability window.

To increase the energy density of LIB anodes, silicon's high specific capacity (3579 mAhg$^{-1}$ for $Li_{15}Si_4$) and relatively low working potential of <0.4 V vs. Li/Li+ [7,12,13] make it a promising contender as either an addition to carbon-based electrodes, or an anode in its own right. However, when cycled in conventional carbonate electrolytes it shows rapid capacity fade which is commonly attributed to large volume expansion during lithiation (>300% for $Li_{15}Si_4$). Successive volumetric changes upon discharge/charge are expected to cause fracture of the SEI and Si pulverisation, leading to repetitive exposure of fresh Si surface and thus ongoing electrolyte decomposition and SEI thickening[14,15]. Nanosizing has proved an effective strategy to accommodate large volume changes in Si without significant pulverisation, thereby reducing the rate of capacity fade[14–17]. However, the increase in surface area leads to a corresponding increase in undesired side reactions, and thus more rapid loss of active Li[18]. One of the most successful strategies employed for suppressing continuous electrolyte decomposition on Si anodes has thus been the use of electrolyte additives. The addition of fluoroethylene carbonate (FEC) has been shown to significantly improve capacity retention[19–21], however its role in SEI stability and improved capacity retention is not fully understood. It has been previously suggested that fluoride ions from reduction of FEC lead to increased LiF formation yielding a more stable SEI[20,22]. Alternatively, the improved performance has been linked to increased formation of particular polymeric species such as polycarbonates or highly cross-linked polyethylene oxide[19,23–25].

Despite the importance of the SEI to LIB performance, detailed mechanistic understanding of its formation and resulting structure remain elusive. Determining how the SEI forms and evolves during battery cycling is challenging due to the complex multistep reactions involved, and the difficulty in directly observing the buried, nm-scale interphase during battery operation[26,27]. Widely employed ex situ surface analysis approaches only capture snapshots of SEI chemistry and are prone to ambiguity due to the need to handle the often highly reactive electrode surfaces in glovebox atmospheres[28], and to rinse away electrolyte residue prior to surface analysis that inevitably disturbs the SEI[29]. Nevertheless, a broad picture has emerged of the SEI consisting of an inner region close to the electrode that is dominated by inorganic components such as LiF, whilst the outer SEI is richer in organic species[26,30–34]. However, questions still remain about when and how this layering occurs. On graphite electrodes it has been suggested that repeated decomposition and reduction reactions lead to layering emerging with ongoing cycling[33], whilst other work has highlighted that the formation potentials of key SEI components such as LiF can vary dramatically for different electrode surfaces[35].

Here we demonstrate *operando* soft X-ray absorption spectroscopy (sXAS) measured in interface-sensitive TEY mode as a powerful technique for probing both organic and inorganic components during SEI formation. This allows access to light elements forming at buried electrode-electrolyte interfaces, thanks to the μm-scale penetration of the incoming X-rays combined with the nm-scale interface sensitivity of photo-generated electrons crossing this interface. We thereby reveal potential-dependent chemical changes occurring on amorphous Si (a-Si) anode surfaces in conventional carbonate-based electrolytes with and without FEC present as an additive. Using an enclosed *operando* cell design, representative performance is achieved during electrochemical cycling, with O, F, and Si K-edge absorption spectra acquired under applied bias. This avoids the loss of important information and potential ambiguity introduced as a consequence of the more common post-mortem style measurement approaches. Cross-referencing to cycling data, complementary bulk sensitive fluorescent yield (FY) XAS measurements, and density function theory (DFT) calculated spectra enables identification of the electrolyte and SEI species, and the dominant mechanisms of SEI formation. Without FEC present, LiF formation is detected at 0.6 V vs. Li/Li+ prior to significant lithiation of the a-Si, whilst at lower potentials the SEI grows in thickness with an increased contribution from organic components containing -C(=O)O- species. The observed sequential formation of inorganic and organic components is implicated in layering of the SEI. With FEC as an additive we see the onset of SEI formation at much higher potentials (1.0 V vs. Li/Li+), and attribute the improved cycle life seen with this additive to the rapid healing of SEI defects formed during delithiation.

## Results and discussion

### Electrochemistry and surface morphology

Figure 1a compares electrochemical data for a-Si films cycled in half cell configurations with and without FEC present in the electrolyte. The cells were cycled between 0.1 and 2 V at C/30 for the first cycle and C/10 for subsequent cycles. Both the LP30 and LP30+FEC cells show a coulombic efficiency of ~70% during the first cycle that recovers to ~98% within a few cycles. This is attributable to irreversible restructuring of the a-Si, surface $SiO_2$ reduction, and electrolyte decomposition related to formation of the SEI during the first discharge/lithiation. Subsequent cycles show charge/delithiation capacities of ~2800 mAhg$^{-1}$ consistent with the theoretical capacity of Si (3579 mAhg$^{-1}$) and the restricted lower cut-off potential of 0.1 V. After ~50 cycles, both the efficiency and capacity of the cell containing only LP30 electrolyte start to drop, with the capacity falling below 1000 mAhg$^{-1}$ by cycle 100. For the cell with the FEC additive, the coulombic efficiency and rate of capacity fade remain relatively unchanged over 100 cycles, clearly demonstrating the stabilising effect of the FEC additive.

Figure 1b displays the voltage-capacity curves for the first cycle of the cells shown in Fig. 1a, in addition to cells where the lower cut-off potential was extended to 0.005 V. The corresponding differential capacity curves are shown in the top panel of Fig. 1c. Increased discharge and charge capacities are observed when cycling to 5 mV (~5300 mAhg$^{-1}$ and ~3500 mAhg$^{-1}$) compared to 100 mV (~4100 mAhg$^{-1}$ and ~2700 mAhg$^{-1}$), consistent with the greater extent of lithiation expected at lower potentials, as well as more irreversible capacity associated with side-reactions. All cells show similar voltage profiles, with a relatively steep drop from open circuit voltage to ~0.3 V on lithiation, followed by a flat plateau and then gradual reduction in voltage to the cut-off potential. The flat plateau is seen as a sharp peak at ~0.31 V in plots of differential capacity, and is attributable to the initial lithiation step in a-Si, which has more facile lithiation pathways compared to crystalline Si[13,36,37]. This peak is not observed in subsequent cycles, and can be considered as an irreversible reorganisation of the Si structure in order to form an amorphous silicide. Two broader peaks are apparent at ~0.23 V and ~0.10 V, with the lower potential peak being slightly cut off in the cells cycled to 0.1 V for obvious reasons. The origins of these have been discussed in detail previously, and are attributable to phase transitions of a-$Li_xSi$ with increasing x[38–42]. The peak at ~0.23 V represents the formation of a silicide with x ≤ 2 (upto a-$Li_2Si$), while the peak at ~0.10 V is related to the formation of a silicide with a stoichiometry ratio of 2 ≤ x ≤ 3.5 (upto a-$Li_{3.5}Si$). Two broader peaks are seen on delithiation, one at ~0.27 V and the other at ~0.47 V. These are the reverse phase transitions, with that at ~0.27 V representing $Li_{3.5}Si \rightarrow Li_2Si$, and the ~0.47 V transition of $Li_2Si \rightarrow$ a-Si.

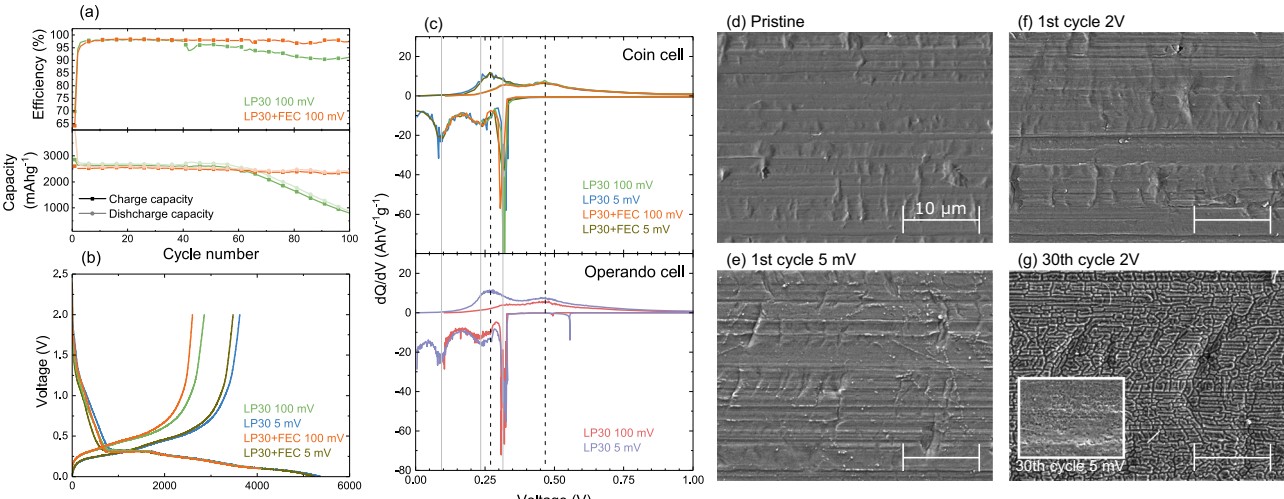

**Fig. 1 | Electrochemical and structural characterisation of a-Si electrodes.**
**a** Electrochemical performance of two coin cells with a-Si(100 nm) thin-film anodes on Ni-coated Cu, containing either LP30 or LP30+FEC electrolytes, showing differences in coulombic efficiency (top) and charge and discharge capacities (bottom). **b** Voltage profiles for the first cycle of the coin cells containing LP30 and LP30+FEC electrolytes cycled to 100 mV and 5 mV. **c** Differential capacity plot (dQ/dV) for the first cycle of the coin cells containing LP30 and LP30+FEC electrolytes cycled to 100 mV and 5 mV (top), and of *operando* cells containing LP30 electrolyte cycled to 100 mV and 5 mV (bottom). We note that the active material mass used for the *operando* cell is based on the areal density of the a-Si electrode and the area in contact with the electrolyte as defined by a viton O-ring (see Fig. 2). **d–g** Scanning electron micrographs of pristine a-Si(100 nm) on Ni-coated Cu (**d**), after cycling to 5 mV (**e**) and then to 2 V (**f**) during the 1st cycle, and after the cycling to 2 V (**g**) during the 30th cycle using LP30 electrolyte with no additive. The inset of (**g**) shows an electrode after cycling to 5 mV during the 30th cycle. All micrographs including the inset are shown at the same scale to aid direct comparison.

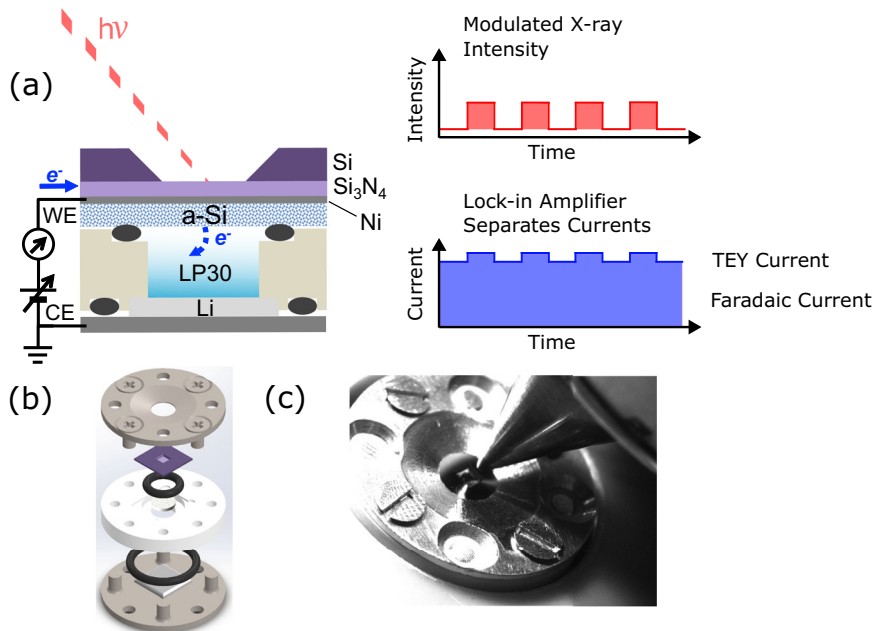

**Fig. 2 | *Operando* XAS cell construction and working principle. a** Cross-sectional schematic of the electrochemical cell for *operando* XAS, showing the a-Si(20 nm)/Ni(20 nm) and Li electrodes and the LP30 electrolyte. A $Si_3N_4$ (100 nm) membrane is used as the X-ray transparent window and the electrodes are connected to an external circuit allowing *operando* cycling. TEY-XAS is enabled by measuring the current generated under X-ray illumination, employing a beam-chopper and lock-in amplifier set-up to separate faradaic and TEY currents. **b** Deconstructed 3D cell design, showing the positions of the O-rings, flanges, membrane, Li electrode and separator. **c** Photograph of *operando* cell in the experimental setup.

We note that lithiation at potentials <50 mV can potentially lead to the formation of the metastable, crystalline $Li_{15}Si_4$ phase, which has been associated with accelerated failure of the Si anode[13,36,38]. The delithiation of this phase exhibits a characteristic sharp feature at ~0.42 V in plots of differential capacity, and as this is absent from the data herein we can exclude the involvement of $Li_{15}Si_4$. This suppression of undesired $Li_{15}Si_4$ formation is attributable to a lack of low-potential holds as well as the mechanical confinement of the Si as a result of its strong adhesion to Ni, which motivates the selected cycling protocol and choice of Ni as current collector[36].

Figure 1d–g shows SEM of a-Si cells cycled in LP30 electrolyte (no additive) and stopped at different stages of cycling revealing changes in the electrode morphology. Initially, the pristine a-Si electrode (Fig. 1d) appears relatively smooth, showing topography that matches the

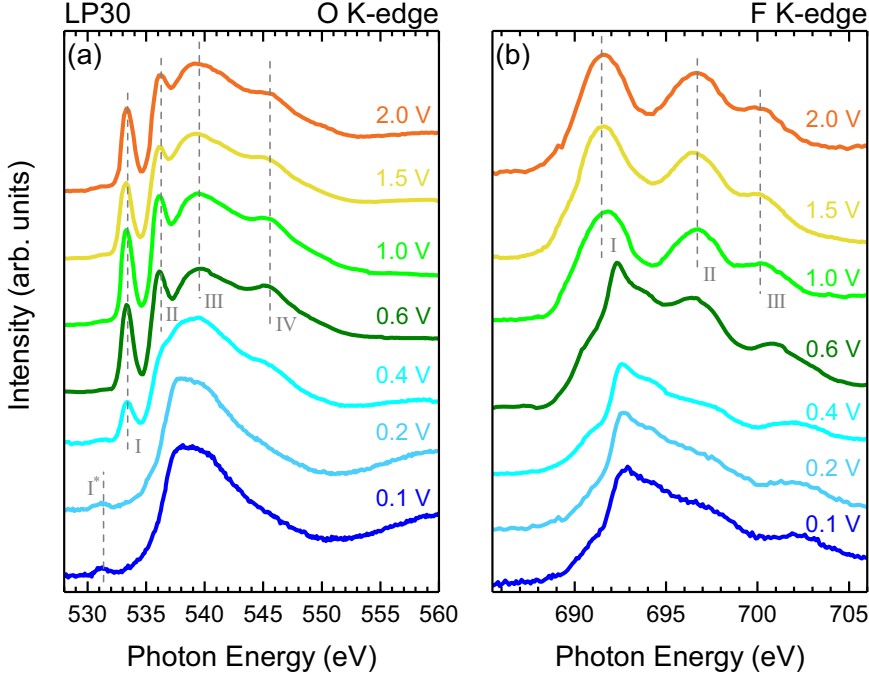

**Fig. 3 | *Operando* TEY-XAS without any additives.** TEY XAS of (**a**) the O K-edge and (**b**) F K-edge taken during cycling at different potentials ranging from 2.0 V down to 0.1 V, for a cell containing LP30 electrolyte without any additives.

rolling striations of the underlying Cu substrate. On cycling to 5 mV during the 1st cycle (Fig. 1e), the electrode surface remains similarly smooth despite the large expected volume increase due to lithiation of the a-Si, indicating this is primarily accommodated through swelling of the electrode thickness. On cycling back to 2 V (Fig. 1f), the overall surface morphology remains largely unchanged although a small amount of cracking can be discerned close to distinct topographic features such as striations. Therefore, during the first cycle the thin film electrode remains continuous, with SEI formation expected to occur predominantly at the exposed electrode surface rather than penetrating through the electrode thickness. Close inspection of Fig. 1e reveals several small bright dots << 1 μm in lateral dimensions, however these are not as apparent in Fig. 1f. These are likely products formed during cell disassembly and inert transfer, reflecting the high reactivity of the lithiated a-Si. Following more extended cycling (30 cycles, Fig. 1g) much larger morphological changes are apparent with an interconnected network of cracks apparent at 2 V. This is attributable to contraction of the lithiated a-Si as it is delithiated at high potentials leading to the formation of silicon islands of <1 μm in at least one direction, which are separated by sizable gaps of ~200 nm. These provide pathways for electrolyte to penetrate and form fresh SEI through the electrode thickness. These cracks are seen to be refilled to some extent through expansion of the a-Si when it is again lithiated (inset of Fig. 1g). This repeated cracking and SEI formation eventually leads to isolation and/ or delamination of Si islands from the current collector, contributing to the capacity fade observed after repeated cycling.

Whilst lithiation of a-Si is clearly distinguishable in the electrochemical data of Fig. 1 and the associated morphological changes revealed by SEM, only limited information on the SEI formation process is directly apparent. The capacity seen on first discharge for voltages >0.3 V in Fig. 1b is commonly attributed to SEI formation, with FEC and EC decomposition processes expected at ~1.2 and ~0.8 V respectively[24]. However, the wide variety of possible reaction pathways and products are challenging to distinguish from electrochemical analysis alone. Therefore, in this study we implement an *operando* cell for TEY-XAS to further reveal the chemical processes occurring at the electrode-electrolyte interface during SEI formation.

## *Operando* X-ray absorption spectroscopy

Figure 2 illustrates the construction of the *operando* XAS cell, including the principle of X-ray modulation required to separate Faradaic and TEY currents. This is similar to other designs for sXAS of liquids[43–46], where a suspended $Si_3N_4$ membrane (~100 nm thick, 0.5 × 0.5 mm or 1.0 × 1.0 mm window) forms an X-ray transparent and pressure resistant window, with a Si frame (~200 μm thick, 5 × 5 mm frame) acting as a support for the window. However the design has been adapted to more closely resemble the coin cell geometry widely used in battery research, allowing controlled application of pressure between the electrodes during assembly.

The bottom panel of Fig. 1c confirms that the *operando* cell displays representative electrochemistry directly comparable to coin cell measurements using the same cycling protocols and a-Si(100 nm) electrode films (see Supplementary Fig. S1 for the corresponding voltage profiles). The dQ/dV plots for the coin cells and *operando* cells are very similar with the same key features present. The alignment of these with voltage is excellent confirming no significant additional overpotential in the *operando* cell, and the shapes and areas of the peaks are also in good agreement.

Figure 3 displays the results of *operando* TEY-XAS measurements performed using the cell assembly seen in Fig. 2a, b, with LP30 electrolyte without any additives. Both the O K-edge (Fig. 3a) and F K-edge (Fig. 3b) spectra were acquired during several fixed potential holds (2.0, 1.5, 1.0, 0.5, 0.4, 0.2, 0.1 V) which are reached by galvanostatic cycling between them at a rate of C/20. In this way the SEI formation can be tracked with a total first-cycle duration of ~30 hrs, aligning with the C/30 cycling rate used for the electrochemical data presented in Fig. 1. Although the focus herein is the SEI components, Si K-edge measurements were also performed at several potentials confirming removal of $SiO_2$ from the Si surface and the lithiation of Si at low potentials (see Supplementary Fig. S2), as expected from the cycling data presented in Fig. 1.

We first consider the evolution of the O K-edge spectra (Fig. 3a). Upon cycling to 2.0 V the O K-edge lineshape displays a number of features (peaks I–IV in Fig. 3a). Strong narrow peaks are seen at ~533.4 eV (peak I), and ~536.3 eV (peak II), in addition to broader

features at ~539.3 eV (peak III) and ~545.4 eV peak (IV). These resemble the main features of the O K-edge spectra reported for LiBF$_4$ in propylene carbonate (PC), whose oxygen environments are similar to those of EC[47], and for LiPF$_6$ in both DMC and EC/DMC[48], as well as X-ray Raman measurements of LiPF$_6$ in both PC and EC/DMC[49]. We therefore attribute these features to the electrolyte solvents EC/DMC.

As the cell is cycled to progressively lower potentials, the electrolyte features persist relatively unchanged until 0.4 V is reached. At this point the intensity of these features diminishes, although peak I is still clearly visible (with reduced peak intensity). At even lower potentials, the electrolyte peaks are no longer discernable and the features above 534 eV are similar to those of the as-deposited a-Si (see Supplementary Fig. S3), with the main peak maximum found at ~538.0 eV and a post-peak at ~540.0 eV. This lineshape closely resembles SiO$_2$[50,51] or possibly oxidized Si$_3$N$_4$[52]. We also performed measurements of as-deposited a-Si/Ni/Cu (as used in the coin cells), and a-Si/Ni/Si$_3$N$_4$ and Ni/Si$_3$N$_4$ membranes which all show the features of SiO$_2$ in their O and Si K-edges (see Supplementary Fig. S3). We note the spectrum also resembles measurements reported using similar *operando* techniques through Si$_3$N$_4$ membranes, but which probed aqueous systems and attributed the line shape to interfacial water[44,53]. However, for the spectra presented herein, where aprotic solvents are used, we can exclude any significant contribution from interfacial water. Furthermore, we note that the spacing between main-peak and post-peak is expected to be ~2 eV larger for water than seen in the spectra herein. A small additional feature is seen at lower energy (~531.5 eV) which we attribute to the O 1s to $\pi^*$ transition of carbonyl (C=O) groups. This peak is seen to grow in intensity between 0.4 and 0.1 V. As we discuss further below, this is attributable to the growth of organic components of the SEI.

The F K-edge spectrum at 2.0 V (see Fig. 3b) shows three clearly distinguishable peaks centred at ~691.7 eV (peak I), ~696.7 eV (peak II) and 700.3 eV (peak III), with a fourth (peak IV) at much higher energy (~714.2 eV - see Fig. 4). This corresponds closely to the expected lineshape for PF$_6^-$ ions[54–56], and so we assign this spectrum as such, consistent with others seen in the literature[57,58]. Note that reference LiPF$_6$ spectra reported in refs. 57, 58 show a pronounced low energy shoulder not seen in the *operando* measurements reported here, which may correspond to contamination or degradation of the ex situ reference sample, highlighting a potential benefit of the *operando* cell methodology. Upon reaching 0.6 V another significant change in line shape is seen, with the peaks related to LiPF$_6$ diminishing and the emergence of a spectral shape that is attributable to LiF[59–63] (see Supplementary Fig. S4 for LiF reference spectrum). Interestingly, the onset of this change occurs at higher potential than the changes seen in the O K-edge (Fig. 3a), and is attributable to the reduction of residual HF present in LiPF$_6$-containing electrolytes[35,64]. The LiF features become increasingly prominent with cycling to lower potentials and LiPF$_6$ features are no longer discernible below 0.4 V.

In order to further understand the electronic structure that gives rise to the observed XAS lineshapes, we use DFT calculations to simulate spectra of the electrolyte components. Figure 4a shows the measured O K-edge at a potential of 1.0 V (top), and the combined simulated spectra for isolated EC and DMC molecules (bottom). The salient spectral features are labelled (I–IV) to assist in comparison. The theoretically determined spectrum matches the experiment reasonably well, displaying a similar intensity profile and number of features, but being more contracted in energy, which is attributable to the imperfect description of the exchange-correlation functional chosen. Note that the features are broader in the experimental data, but display more structure in the simulated spectra (especially peak III). This is related to the isolated molecule description used here to model the system. It has been shown that employing molecular dynamics to describe the forces between atoms in denser systems can broaden some features in the spectrum[65,66]. However, this is not considered

here as it is not expected to provide further insight relevant to the experimental data obtained.

The EC/DMC peaks can be understood using a simple molecular orbital approach (similar to that used in ref. 47, see Supplementary Fig. S6 for the projected isosurfaces displaying the bonding character of the atoms), where the two narrow peaks (I and II) are related to transitions to the $\pi^*$-antibonding orbitals from the O 1s of the C=O (EC 3 and DMC 3) and C-O-C (EC 1+2 and DMC 1+2) bonding configurations respectively. DMC and EC exhibit similar bond lengths for both their C=O and their alkoxy/ring C-O bonds (see Supplementary Table S1) and so it is unsurprising that these states overlap in energy. The broad features (III and IV) are related to transitions into the $\sigma^*$-antibonding states, at higher energies than the $\pi^*$ orbitals. The $\sigma^*$-derived peaks mainly arise from transitions from the O 1s of the alkoxy/ring oxygen, but there is some spectral overlap with transitions from the O 1s of the carbonyl oxygens of both EC and DMC, indicating multiple contributions. Interestingly, DMC does not appear to contribute significantly to peak IV, which mainly arises from the ring oxygen in EC. Instead, between the cyclic and linear carbonate we see intensity being redistributed to lower energies, contributing more significantly to peak III, which may relate to the slight bond contraction of the C-O bonds and elongation of the C=O bond. This is further supported by the projected isosurfaces in Supplementary Fig. S6, which show very low electron density around the core-hole alkoxy oxygen atom of the DMC compared to that of the EC. We note that the calculations performed neglect the effects of molecular interactions, which distort the molecules and cause a further redistribution of states and greater spectral broadening[47].

Figure 4b shows the measured F K-edge at a potential of 2.0 V (top), and the simulated spectrum of an infinite LiPF$_6$ crystal including a core-hole on the F atom (bottom). Core-hole spectra for both LiPF$_6$ and PF$_6^-$ were calculated (see Supplementary Fig. S7), and found to be very similar in spectral shape. The simulated spectrum again matches the experimental data reasonably well, displaying a similar intensity profile and number of features. However, in contrast to the EC/DMC simulation, where both the Gaussian and Lorentzian components of the theoretical spectral broadening were set to 0.5 eV, a much greater Lorentzian broadening was required to better match the experiment (1.2 eV broadening was applied). In addition, we find excellent agreement between the valence partial density of states obtained from the DFT calculations and valence band X-ray photoemission spectra (from Dedryvère et al.[67]) seen in Supplementary Fig. S8, giving confidence in the applicability of the simulation. Given the alignment between the simulated spectra and experimental reports in literature, the spectral features in Fig. 3b at 2.0–1.0 V can be confidently assigned to PF$_6^-$ ions in the electrolyte. The electronic structure that gives rise to these features can be attributed based on a simple molecular orbital description of octahedral coordinated molecules, with peak I corresponding to the F 1s transitions into unoccupied F($p$)-P($s$) hybridized orbitals ($a_{1g}$ symmetry), peak II to transitions into F($p$)-P($p$) hybridized orbitals ($t_{1u}$ symmetry) and peak III to transitions into unoccupied F($p$)-P($d$)-hybridized orbitals ($t_{2g}$ symmetry)[55,68]. The $e_g$ band is found at even higher energy.

Figure 4c shows the measured F K-edge at a potential of 0.1 V (top), and the simulated spectrum of an infinite LiF crystal (bottom). These appear extremely similar, with features I–IV matching very closely. An additional shoulder I* is seen in the experiment (~690.9 eV) but not in the theory. This is seen even more strongly in Supplementary Fig. S4. We attribute this feature to excitonic effects[59,60], which are not captured in the DFT calculations.

## Spectral evolution

Having assigned the chemical species and how their electronic structure gives rise to the features seen in Fig. 3, we now rationalise the observed spectral evolution with potential. It is first instructive to

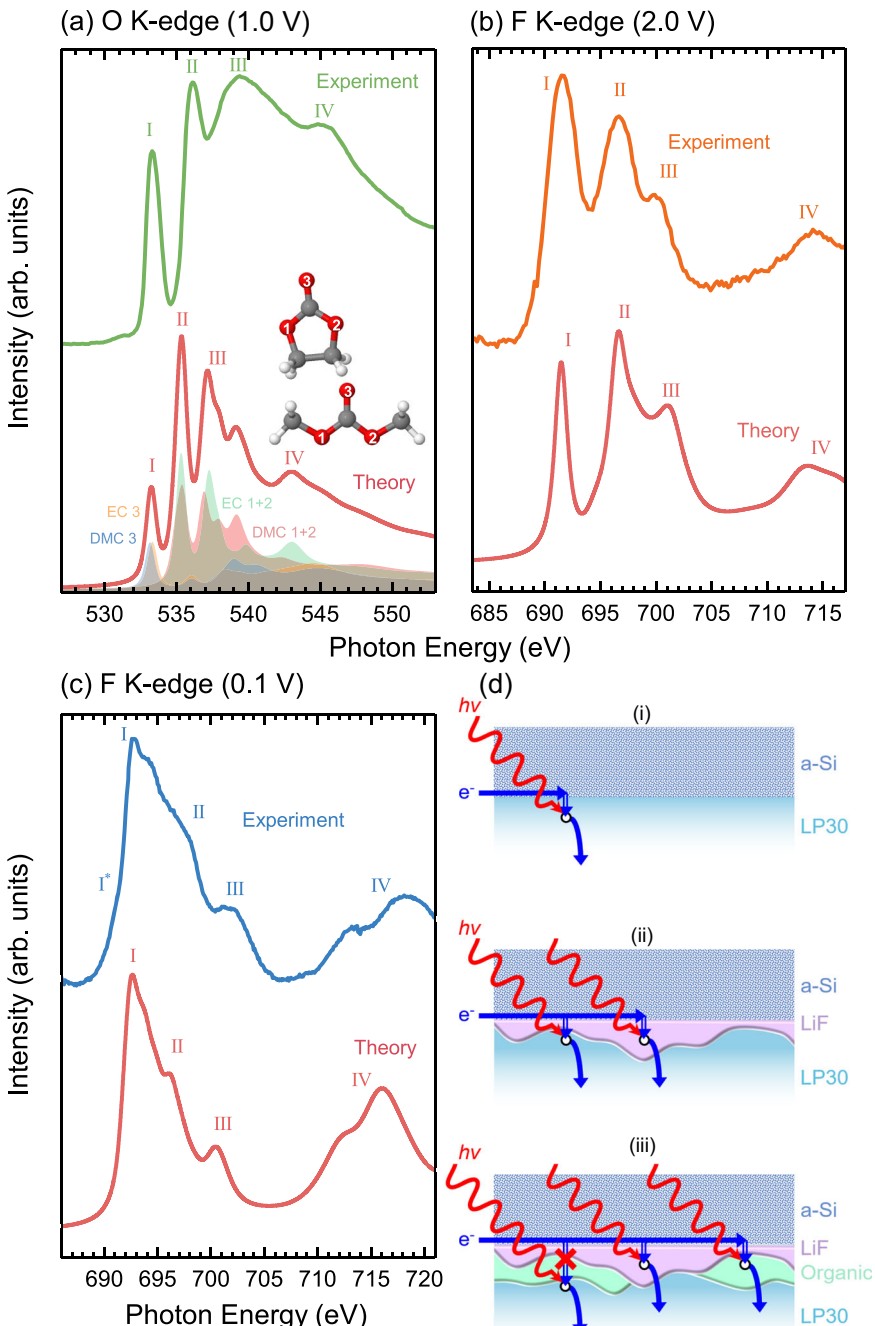

**Fig. 4 | Simulated and experimental X-ray absorption spectra. a** O K-edge for the the addition of EC and DMC isolated molecules (bottom) and the experimental data (top) recorded at 1 V bias. **b** F K-edge simulated for an LiPF$_6$ infinite crystal (bottom) and experimental data (top) recorded at 2.0 V bias. **c** F K-edge simulated for a LiF infinite crystal (bottom) and experimental data (top) recorded at 0.1 V bias. **d** Schematic representation of the different paths that electrons take in the cell after photoexcitation.

consider from where the TEY-XAS signal arises. TEY-XAS detection relies on (1) creation of a core-hole by X-ray absorption, (2) emission of Auger electrons and secondary electrons produced by relaxation and inelastic scattering respectively, and (3) an electron conductive path to replenish the emitted electrons. The X-ray attenuation length provides a measure of the depth from an interface over which X-rays are absorbed, and core holes created: typically a few hundred nm up to several $\mu$m for the core levels and materials considered here. Auger emission is thus expected throughout this depth, but only those Auger and secondary electrons that escape across the electrode-electrolyte interface will contribute to the TEY signal. For electronically conductive materials, the range over which the Auger electrons

inelastically scatter largely defines the maximum probing depth, giving rise to the interface sensitivity associated with TEY detection (typically 5–10 nm for absorption edges of 500–800 eV)[69,70]. However, when the TEY signal arises from an insulating layer on top of an electronic conductor, the need for a conductive path (i.e. tunneling) can become dominant in determining the interface sensitivity. As long as the insulator is thinner than the Auger electron escape range, the shorter range of electron tunneling means the detected TEY signal is most sensitive to species close to the buried interface with the electronic conductor.

Typically for TEY-XAS measurements in vacuum, either the sample is grounded and electrons emitted under X-ray illumination are

collected by an isolated counter electrode (CE) connected to ground through a current amplifier, or the sample is connected to ground through the current amplifier and the CE grounded (chamber walls often fulfil this role). For the *operando* TEY-XAS performed herein, the a-Si working electrode (WE) is connected to a current amplifier whose input is biased relative to the Li counter electrode (CE) to have the same voltage as the cell (see Fig. 2a). We note that this is essentially the same detection method as has been recently referred to as 'total ion yield' (TIY) XAS[71,72], albeit with a voltage applied between the electrodes of the current amplifier to avoid short-circuiting the cell. However, we agree with the recent article by van Spronsen et al[73]. that this is not a bulk sensitive measure, and thus refer to it as TEY-XAS as in previous studies[44,46,53]. Although X-ray ionisation occurs within the bulk of the electrolyte, the short range over which the electrons scatter will mean they quickly recombine with ions such that local charge neutrality is maintained within the electrolyte. Only electron/ions that cross the illuminated interface of the WE with the electrolyte will contribute to the collected TEY signal, with charge neutrality of the electrolyte maintained by a corresponding charge transfer at the CE.

Prior to the onset of SEI formation, the O and F K-edges are dominated by features corresponding to the EC/DMC and solvated $PF_6^-$ ions (2.0–1.0 V in Fig. 3). Figure 4d-i shows the expected detection mechanism, with X-ray absorption creating holes which are replenished by the a-Si electrode if sufficiently close, giving an interface sensitive signal corresponding to the electrolyte components. At 0.6 V the O K-edge remains essentially unchanged, displaying the components expected for the EC/DMC solvent with the same intensity profile. However, the F K-edge now shows features of both $LiPF_6$ and LiF, indicating LiF formation in the early stages of SEI formation. Figure 4d-ii shows how both LiF and the electrolyte components can be simultaneously detected, on the basis that there are regions of the LiF thin enough for electrons supplied by the a-Si to tunnel through to replenish photo-generated holes in the electrolyte close to the electrode interface.

It is not until the applied bias reaches 0.4 V that the O K-edge begins to evolve significantly, with the attenuation of the electrolyte features in both the O and F k-edges, and emergence of organic decomposition products in the O K-edge (peak I*). The electrolyte components are completely attenuated below -0.2 V, whilst LiF remains detectable, however with an decreased signal-to-noise ratio. As shown in Fig. 4d-iii, the absence of the electrolyte features, indicates growth of an organic layer and possible thickening of the LiF that suppresses electron tunneling from the a-Si into the electrolyte. We note that this disappearance of the electrolyte signal corresponds with a key function of the SEI, to provide an insulating layer that prevents ongoing electrolyte decomposition. A weakened LiF signal seen at 0.1 V is also consistent with overgrowth of the organic layer and/or LiF thickening, but indicates that this remains below the range over which the Auger electrons inelastically scatter such that Auger-generated secondary electrons can still transfer to the electrolyte (see Fig. 4d-iii). Similarly the organic component in the O K-edge (peak I*) remains detectable at regions where the LiF is sufficiently thin for tunneling to occur and where the organic layer is thin enough for Auger-generated secondary electrons to transfer to the electrolyte. We note that the Si K-edge can be measured across the whole potential range, consistent with electron tunneling not being required for its detection. Also note the longer range over which Auger electrons inelastically scatter for this higher energy edge (See Supplementary Fig. S2 for detection mechanisms of a-Si). The spectral evolution observed is consistent with the inner SEI (closest to the a-Si) being rich in LiF, while the outer SEI contains more organic species[26,30–34]. It further indicates that this layering exists from the first SEI formation cycle, and is the result of LiF deposition at higher potentials followed by organic components as the potential is lowered further, rather than emerging only as a result of repeated decomposition and reduction reactions during ageing.

In addition to TEY-XAS, more bulk sensitive *operando* FY-XAS was also performed, that although less sensitive to the evolution of interfacial species (~1 μm attenuation length compared to <10 nm maximum probing depth in TEY[69,70]), allows simultaneous probing of the electrolyte throughout SEI formation. The same LP30 electrolyte was used with a model Ni(20 nm) WE, with the absence of a-Si minimising signal distortion and self-absorption associated with thickening of the electrode due to lithiation. In addition, the negligible capacity of Ni allows cycling between potentials at a faster rate, with a 15 min hold at each potential in this case.

In contrast to the TEY-XAS, features of LiF are apparent in the F K-edge at OCV. We note that LiF formation through electrocalytic reduction of residual HF in $LiPF_6$-containing electrolytes is expected to occur at much higher potentials (~2 V vs Li⁺/Li) on Ni surfaces compared to the -0.6 V vs Li⁺/Li seen for a-Si in Fig. 3b[35,64]. We cannot exclude that increased radiolytic electrolyte decomposition may also lead to earlier onset of LiF formation. However we note that repeated measurements of the O K-edge at OCV show no significant changes over time, suggesting any beam-induced SEI growth is limited. Either way, the radiolytic decomposition products of carbonate-based electrolytes are expected to closely resemble those due to electrochemical reduction[74–79].

Figure 5a shows that the four main peaks (I–IV) associated with the EC/DMC solvent are apparent in the FY spectrum of the O K-edge at OCV. On cycling to lower potentials, little variation from the spectrum at OCV is seen until 0.7 V, where a low energy shoulder corresponding to peak I* begins to grow in intensity, becoming clearly apparent at 0.05 V. Alongside this, peak I is seen to decrease in intensity while peak III grows, with peaks II and IV maintaining similar intensities. There is also some intensity gain apparent at around 535 eV, leading to a less-pronounced minimum. We demonstrate that the changes seen do not simply arise from changes in self-absorption/saturation effects due to ion rearrangement under applied potential, as when the bias is removed from the cell the spectral shape remains unchanged (see Supplementary Fig. S9). Note that changes in spectral shape due to the geometrical effects of self-absorption and saturation related measurements in FY mode[80–82] are complex in this system due to the measured signal coming from multiple layers of different thicknesses and densities, including those growing electrochemically during the measurement. This of course makes any correction schemes, which are nearly always based on some simplifying assumptions regarding the sample geometry, very difficult to implement[83–86]. However, whilst we do not claim this data is free from self-absorption or saturation effects, assuming smoothly varying absorption coefficients (apart form at the step edge)[87] and noting that the measurement geometry stays fixed throughout the experiment and that the SEI layer is much thinner than the X-ray attenuation length, we can expect a uniform effect across the spectra, i.e. contiguous features of similar intensity should not both grow and shrink across the energy range. Hence, whilst the FY spectra can't be assumed to accurately map the absorption coefficient across the O K-edge, the changes seen are consistent with chemical changes rather than geometric ones.

The growth of peak I* is consistent with formation of organic SEI components, attributed earlier to transitions from the O 1s of C=O to π*-antibonding orbitals. Figure 5b plots the associated transition energies for a variety of organic components as reported in literature, where a distinct trend in transition energy allows them to be grouped based on molecular motif[88]. On this basis peak I* at ~531.5 eV could be attributable to aldehyde (-C(=O)H), ketone (-C(=O)R), carboxyl (-C(=O)OH), and/or ester (-C(=O)OR) groups which have previously been identified in literature[24,33,89–92].

The reduction in intensity of peak I corresponds to attenuation of the C=O related to EC/DMC, consistent with SEI formation involving decomposition of these carbonate species to yield -C(=O) or -C(=O)O-containing motifs that contribute to peak I*. We note that this does not

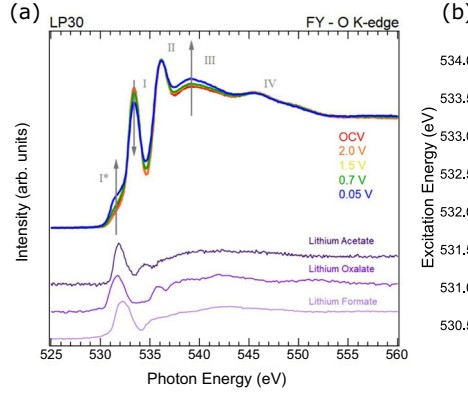
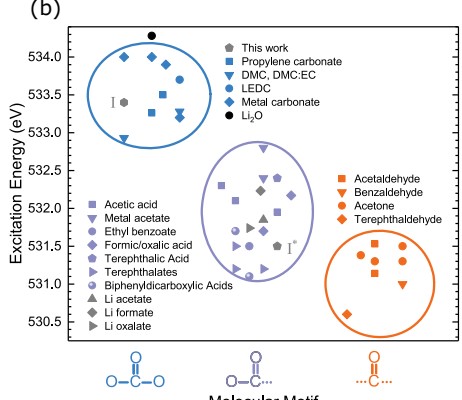

**Fig. 5 | Comparison of Operando O K-edge XAS with various C=O containing reference compounds. a** FY-XAS of the O K-edge taken during cycling at different potentials ranging from OCV down to 0.05 V. Reference spectra of materials sharing the -C(=O)O- structural motif are also shown for comparison. **b** Excitation energies of the O 1s → π* transition for a number of molecules possessing an C=O bonding configuration and either two, one or no C-O bonds[47,49,88,94,98,100,121–132]. Peak positions from (**a**) are included in grey, and $Li_2O$ is also included in black for reference but doesn't correspond to the molecular motifs shown. See Supplementary Table S3 for further information on the data used here.

exclude formation of other carbonate species e.g. lithium ethyl carbonate (LEC), lithium methyl carbonate (LMC), lithium ethylene monocarbonate (LEMC), lithium ethylene dicarbonate (LEDC), lithium butylene dicarbonate(LBDC). However, these are challenging to distinguish here due to significant spectral overlap with EC/DMC (see Fig. 5b). Significantly, a similar attenuation of peaks II-IV is not observed, and indeed peak III is seen to grow in intensity. This suggests that the organic SEI components formed by electrolyte decomposition contribute to these peaks, counteracting the attenuation of electrolyte species by the SEI formation and leading to the growth in peak III.

Aldehyde and ketone groups show only weak features above 535 eV, thus although they may contribute to peak I* they cannot account for the growth in peak III. However, the additional C-O bond in carboxyl and ester groups contributes to additional features in this range. Indeed, recently reported simulations of O K-edge spectra for -C(=O)OH, exhibit features at the appropriate energies[93], with peak I* arising from C=O to π* transitions, C-O to π* transitions contributing to peak II, whilst both C=O and C-O to σ* transitions contribute to peak III, and C=O to σ* to peak IV. Notably, experimental spectra acquired for polymers containing (-(C=O)OCH$_x$) motifs show a close match with the spectral features required, including a close alignment in energy for Peak I*, and additional intensity at peak III[94]. Such motifs would be expected from the reduction of EC or DMC by acyl-oxygen cleavage as previously reported based on gas evolution measurements[95]. We thus consider these the most likely species to be contributing to peak I*. We note that -(C=O)O motifs are also expected to give similar lineshapes and can form by reduction of $CO_2$ generated from electrolyte decomposition, yielding species such as lithium formate, oxalate and succinate[27,96,97]. We include in Fig. 5a FY reference spectra of lithium acetate, lithium oxalate and lithium formate for comparison, all possessing -(C=O)O molecular motifs (see also Supplementary Fig. S9 for TEY-XAS of the same samples). Clearly the lowest energy peaks coincide with peak I* well (see also Fig. 5b). An increase in intensity is also seen above 535 eV in each case, with much of the intensity coinciding with peak III, giving confidence in this motif as the assigned organic component formed. Note that the presence of small contributions from adventitious species cannot be excluded (e.g. in the Li acetate at around ~535 eV[98]). XPS studies of the SEI formed on graphite electrodes, further support the assignment of -(C=O)O- species[34,99].

## FEC additive

We now turn attention to the influence of the FEC additive on the SEI formation. Figure 6 shows the TEY-XAS data recorded using the same cell set-up as previously described, but including 10% FEC in the LP30 electrolyte (EC:DMC:FEC in amounts 45:45:10 wt%). Both the O K-edge and F K-edge show the presence of the same main spectral features, as observed without the FEC (Fig. 3), and the evolution of the spectra proceeds in a similar manner.

The same four main electrolyte peaks seen with LP30 are clearly apparent at 2 V. This lineshape remains largely unchanged down to a potential of 1.0 V, where a rapid change is observed. Two spectra acquired at the hold potential of 1.0 V are displayed, which are separated in time by roughly 15 min (see Supplementary Fig. S10 for rapid scans of peak I performed during this period). The second spectrum shows a clear attenuation of the electrolyte peaks, closely resembling that observed with the LP30 electrolyte at 0.4 V in Fig. 3a. This highlights that a similar evolution is seen but the onset is at a much higher potential when FEC is included as an electrolyte additive. Further changes are seen on cycling to 0.5 V, where although a similar intensity of peak I is retained, the other electrolyte peaks (II–IV) appear more significantly attenuated, with the spectrum above 535 eV now resembling that seen at 0.2 V and below for the LP30 electrolyte. This suggests that peak I no longer corresponds to the EC/DMC electrolyte, but rather decomposition products with a similar XAS feature at the peak I position such as lithium alkyl carbonates (RO(C=O)OLi) e.g. LEC, LMC, LEMC, LEDC[98], and $Li_2CO_3$[100,101]. At 0.4 V, peak I* grows at the expense of peak I but both are still discernible. On further cycling to 0.2 V and below the spectra closely resemble those seen at the same potentials with LP30 in Fig. 3. Given the similarities in their final spectrum, we note that this same evolution may also be occurring with the LP30 electrolyte at lower potentials, but isn't captured at the specific potentials measured.

The F K-edge evolution seen in Fig. 6a also resembles that seen in Fig. 3b, with the three strong resonances of $PF_6^-$ ions in the electrolyte clearly apparent at 1.5 V and above. We see that with the FEC additive included, the spectrum begins to change from the $PF_6^-$ features towards those expected for LiF at 1.0 V, a much higher potential than the 0.6 V for the LP30 electrolyte without FEC. On cycling to lower potentials the LiF lineshape persists, but maintains a noticeably greater signal to noise ratio than seen for F K-edge measurements at similar potentials with LP30 (see Fig. 3a). This may reflect formation of a SEI containing a higher proportion of LiF and/or which is thinner.

The reduction of FEC at higher potentials than EC has been suggested previously based on electrochemical data[24,102,103], but to our knowledge this is the first experimental verification of this using *operando* spectroscopy, allowing us to identify the potentials where

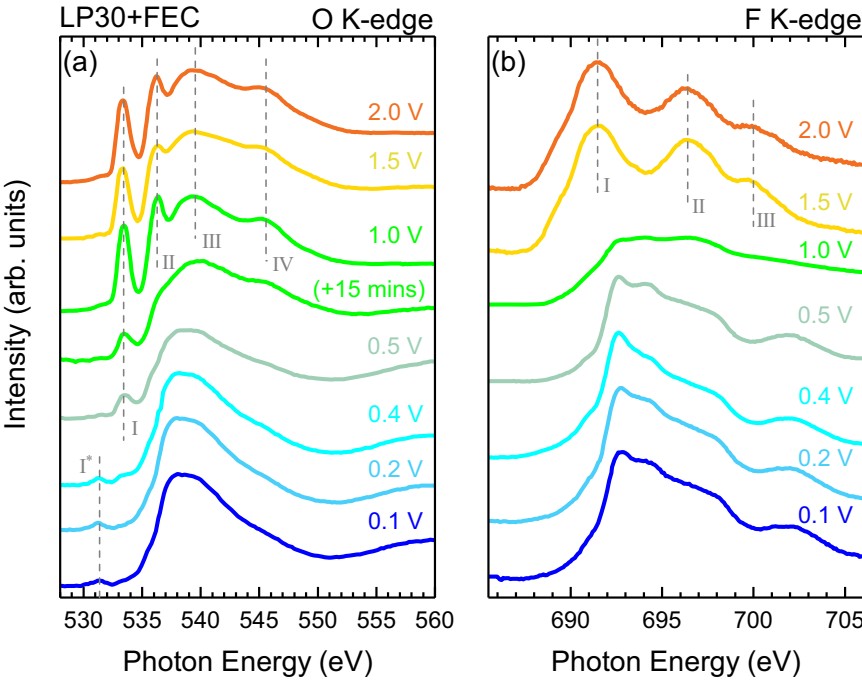

**Fig. 6 | *Operando* TEY-XAS with FEC additive.** TEY XAS of (**a**) the O K-edge and (**b**) F K-edge taken during cycling at different potentials ranging from 2.0 V down to 0.1 V, for a cell containing LP30 electrolyte with FEC as an additive. Note that two O K-edge spectra are displayed at 1.0 V, recorded 15 mins apart from one another.

different chemical changes occur. On this basis we now rationalise the improved cycle life observed for Si electrodes when FEC is used as an additive. As seen in Fig. 1c, lithiation of a-Si occurs predominantly below 0.3 V, and delithiation between 0.2–0.7 V. Significant electrode volume changes are observed in these same potential ranges[104]. This likely leads to mechanical degradation of the SEI and a-Si, and exposure of fresh a-Si/a-Li$_x$Si where additional electrolyte reduction occurs resulting in incremental, irreversible capacity loss[11,105]. Given the SEI forms during the first lithiation cycle, it is expected to be under the greatest mechanical stress towards the end of delithiation due to contraction of the a-Li$_x$Si. Based on the observations herein, for LP30 electrolyte without additives this corresponds to potentials above where significant electrolyte decomposition to form stable SEI components occurs. Thus defect formation and cracking of the SEI can proceed unabated, with freshly exposed electrode surfaces only passivated by new SEI formation during the next lithiation cycle. When FEC is present however, we suggest that the formation of LiF and organic SEI species at higher potentials (up to ~1 V) may act to passivate defects in the SEI as they form, preventing more significant SEI damage and helping to explain the better capacity retention observed in Fig. 1a.

In summary, we demonstrate *operando* sXAS during electrochemical cycling of a-Si anodes in LP30 electrolyte and thereby reveal how SEI formation proceeds with potential with and without FEC present as an additive. Using a modulated X-ray beam and lock-in technique, we are able to extract the TEY signal under applied bias, and thus obtain an interface-sensitive account of the chemical evolution of the SEI. DFT calculations provide insight into the dominant chemical species present in the spectra, and the electronic structure that gives rise to these features. On cyling from OCV, features corresponding to the main electrolyte components (EC, DMC, PF$_6^-$ ions) dominate the O, and F K-edge spectra. LiF formation is seen to begin at a potential of 0.6 V for the LP30 electrolyte. A distinct spectral feature related to organic SEI components containing carbonyl groups emerges at lower potentials (<0.4 V). The spectral evolution observed is consistent with layering of the SEI during formation, with LiF deposition occuring close to the electrode surface at higher potentials, followed by organic components forming on top as the potential is further reduced. The

growth of the SEI is found to electronically isolate the electrode from the electrolyte and thus electrolyte components are no longer detectable in the TEY signal. More bulk sensitive FY measurements are therefore used to assign the organic SEI feature to -C(=O)O-, most likely (-(C=O)OCH$_x$) arising from reduction of EC/DMC by acyl oxygen cleavage. The addition of 10 wt% FEC additive to the electrolyte increases the onset potential of SEI formation, with LiF formation observed at 1.0 V. We suggest that electrolyte decomposition at higher potentials may act to passivate defects in the SEI as they form, helping to account for the significantly improved cycle-life when FEC is present as an additive. Our study of a-Si electrodes herein shows the benefits of combining TEY and FY detection modes under *operando* conditions, with interface-sensitive TEY being well suited to probing the early stages of SEI formation, whilst FY allows simultaneous probing of the SEI and electrolyte throughout formation. We note this same approach can be readily adapted to the study of a wide variety of thin-film anodes and cathodes, as well as different electrolyte formulations, where it can offer new insight into the formation mechanisms of electrode-electrolyte interphases, and their chemical stability.

## Methods

### (a) Electrode deposition and characterisation

Ni thin films (20 nm for *operando* cells and 250 nm for the coin cells) were deposited as current collector and adhesion layer in a custom DC sputter coater (base pressure ≤2.0 × 10$^{-5}$ mbar) using a Ni metal target (PI-KEM, > 99.999%). Following transfer through air, a-Si thin-film electrodes were then deposited on top in a RF sputter coater (CCR Technology, base pressure ≤1.0 × 10$^{-5}$ mbar) using an undoped Si wafer target (PI-KEM, > 99.999%) that was pre-sputtered for 10 mins prior to deposition. The amorphous phase of the Si film was confirmed by Raman spectroscopy (see Fig. S11), using a Reinshaw inVia Raman microscope with backscattering geometry. A laser wavelength of 532 nm was used, with a spot diameter of 1–2 μm and an operating power of 0.2 mW focused through an inverted microscope via a 50 × objective lens. Film thicknesses were determined using a Dektak II profilometer to measure step heights on partially masked pieces of Si wafer, and the mass of active material was determined by weighing of

substrates before and after deposition using a Sartorius Cubis ultra-micro balance. For calculations of gravimetric capacity the average mass difference measured for 12 electrodes was used, yielding $202 \pm 8$ mg/m$^2$ for 100 nm thick a-Si. Scanning electron microscopy (SEM) was performed using a TESCAN MIRA3 FE equipped with a Schottky field emission electron gun and a secondary electron detector. Prior to measurement the coin cells were disassembled in an Ar-filled glove-box ($H_2O < 0.1$ ppm, $O_2 < 0.1$ ppm) and the anodes rinsed with DMC. Samples were then transferred inertly to the microscope vacuum chamber using a Kammrath & Weiss transfer module. Samples were imaged at an acceleration voltage of 5 kV using secondary electron detection.

### (b) Operando XAS cells

Figure 2b shows a deconstructed 3D model of the *operando* XAS cell. This consists of two separated stainless steel flanges, each of which provides electrical contact to one of the electrodes, as required for both electrochemical cycling and detection of the TEY current. The WE is formed by thin-film deposition directly onto the Si$_3$N$_4$ membrane, facilitating X-ray transparency to the electrode-electrolyte interface. There are few restrictions on the thickness of the CE, with a Li foil of a few hundred $\mu$m in thickness used herein. A third flange made from polyether ether ketone (PEEK) is sandwiched between the two steel flanges to ensure electrical isolation of the electrodes. A $\varnothing$ 3 mm hole through the centre of the PEEK flange allows for Li transport, while a $\varnothing$ 3 mm glass-fibre separator (Whatman, Borosilicate, dried at 80 °C in a vacuum oven) placed in this hole prevents electrical shorting. This is saturated with liquid electrolyte during cell assembly, with battery grade LP30 (1 M LiPF6 in EC/DMC 1:1 v/v, Sigma Aldrich, ≤15 ppm of $H_2O$) used as the baseline electrolyte. In some experiments, fluoroethylene carbonate (FEC, Sigma–Aldrich) was included as an additive (10 wt%) in the electrolyte. Cell assembly was performed within an Ar-filled glove-box ($H_2O < 0.1$ ppm, $O_2 < 0.1$ ppm). Viton O-rings are used to ensure a leak-tight seal against the vacuum conditions of the measurement chamber, avoiding evaporation of the liquid electrolyte (see Fig. 2c for a photograph of the cell in situ). The *operando* cells were discharged/charged using a Biologic SP-300 potentiostat.

### (c) Coin cells

Coin cell assembly was carried out in an Ar-filled glovebox ($H_2O < 0.1$ ppm, $O_2 < 0.1$ ppm), using 2032 coin cell components (Cambridge Energy Solutions ltd.), with a-Si(100 nm)/Ni(250 nm) on Cu foil ($\varnothing$ 12.5 mm, 18 $\mu$m thick, > 99.8%) and Li foil ($\varnothing$ 16.0 mm, 250 $\mu$m thick, >99.9%) electrodes, and the same electrolyte (~0.15 mL for each cell) and separator material as used for the *operando* XAS cell. Coin cells were discharged/charged at room temperature and constant current at a C-rate of C/30 (i.e. 120 mA/g of a-Si) for the first cycle using a Biologic VSP, MPG-2, or Lanhe (Wuhan, China) battery test system, and then cycled at C/10 (360 mA/g) for up to 100 cycles (taking ~30 days).

### (d) X-ray spectroscopy

*Operando* XAS of the O and F K-edges was performed in TEY mode at BL 24 (CIRCE), ALBA synchrotron (Barcelona)[106]. To separate the large faradaic current associated with Li$^+$ insertion and SEI formation from the much smaller TEY-XAS current, a motorized rotating slotted disk is used in the path of the beam, which modulates the beam at a selected frequency (1-1000 Hz) as monitored using an optical switch. The drain current from the working electrode (WE) is amplified using a SR570 low-noise current amplifier (Stanford Research Systems) whose input bias relative to the cell's counter electrode (CE) is set to match the cell voltage (see Fig. 2a). A SR830 lock-in amplifier (Stanford Research Systems) is then used to separate the modulated TEY current (<5 nA) from the faradaic current (~250 nA). Further details regarding the experimental set-up have been reported previously[44,46,107]. To our

knowledge this is the first instance of using this beam-chopper and lock-in amplifier set-up to measure a fully constructed Li-ion cell whilst performing *operando* TEY-XAS. Numerous repeat measurements using identical cell set-ups were performed when taking the data presented here, giving confidence in the reproducibility of the behaviour observed. Under X-ray illumination, a slight drop (~10 mV) in the open circuit voltage of the cell is observed which is quickly recovered when the illumination is removed. This corresponds to the small contribution of the photocurrent in discharging the cell. The stability of the materials under X-ray illumination was monitored by repeated measurements of the O K-edge and F K-edges over the course of several hours with no electrochemical current applied. No significant spectral changes were observable, confirming that beam damage effects were negligible on the relevant experimental timescales. Spectra are normalized to the incident photon flux, measured via the photocurrent of a Au mesh or a Au-coated beam-focusing mirror. The photon energy scale is calibrated using the O K-edge of a NiO sample, with its first absorption peak at 532 eV[108–111]. For the O-K edge measurements the photon flux reaching the sample is ~$4 \times 10^{11}$ photon/s, whilst for the F-K edge it is ~$6 \times 10^{11}$ photon/s, with a 50 $\mu$m vertical exit slit size that gives a projected spot size of ~$100 \times 100$ $\mu$m$^2$ parallel to the Si$_3$N$_4$ membrane surface. A photon energy step of 0.1 eV was used for all measurements. The resolving power of the set-up at BL 24 was estimated as $E/\Delta E = 5700$, as obtained from the N 1$s$ vibrational spectrum of N$_2$ gas.

Separate FY-XAS measurements of the O K-edge were acquired using an Al coated Si Photodidode at the I10 beamline, Diamond Light Source (Harwell). The photon flux reaching the sample is ~$1 \times 10^{12}$ photon/s, with a 50 $\mu$m vertical exit slit size used that gives a projected spot size of ~$200 \times 300$ $\mu$m$^2$ parallel to the Si$_3$N$_4$ membrane surface. XAS reference spectra were obtained at beamline B07b (Diamond Light Source) for Lithium acetate (99.9%), lithium formate monohydrate (98%) and lithium oxalate (98%) powders (Sigma Aldrich). All reference spectra were recorded using a verticle exit slit size of 50 $\mu$m. Lithium formate and lithium oxalate were dried on a hotplate in an Ar glovebox at 100 °C for 20 mins before inert transfer into the measurement chamber.

### (e) Theoretical methods

Density functional theory (DFT) calculations were carried out using the plane wave pseudopotential code CASTEP[112]. The Perdew-Burke-Ernzerhof (PBE) form of the generalized gradient approximation (GGA) functional[113] was used for all calculations performed. For calculations of crystalline materials an appropriate unit cell size was chosen to take advantage of the crystal symmetry, while molecular systems were simply placed in a box. Before generating spectra, each structure was geometry optimised. Appropriate plane wave cut-off energies and k-point spacing values were determined via convergence testing of both the total energy and bond lengths (see Supplementary Table S1 and associated Fig. S5). The geometry of the system was considered optimized when the maximum forces on the ions were below 0.01 eV/Å. The geometry optimised structures presented all had bond lengths with a maximum deviation of 2% from experimentally determined values, most displaying <1%.

After geometry optimisation, core-hole spectral calculations were performed[114,115]. Since core orbitals are not treated explicitly in the pseudopotential method, a unique pseudopotential is generated for an excited atom possessing a core-hole. Here we focus on the O and F K-edges (transitions from the 1$s \rightarrow 2p$ levels) so that the core-hole is placed on the 1$s$ orbital. For the case of molecular systems, the cells used were sufficiently large to prevent neighboring excited potentials interacting with each other. However, for crystalline materials we generate a supercell to prevent this. For the spectral calculations, increasing the plane wave energy cut-off, k-point sampling (for both the generation of the density and spectrum) and distance between core-hole atoms was found to have no visible effect on the spectrum

(see Supplementary Table S2 for calculation parameters used). Spectral calculations were handled using Optados[116]. Gaussian (representing instrumental broadening effects) and Lorentzian (representing the state's lifetime) broadening was performed using full widths at half maximum of 0.5 eV for both Gaussian and Lorentzian components in EC, DMC and LiF, while the Lorentzian component was increased to 1.2 eV for $LiPF_6$. We note here that previous studies suggest the Lorentzian width for both O and F related atomic systems to be on the order of 0.1–0.2 eV[117,118], and for the Gaussian components to have widths of only ~0.1 eV according to the relation for the resolving power of BL 24 discussed above. However, the experimental data was found to be much broader than this, which we can partially attribute to the effects of molecular dynamics for non-isolated molecular systems. Other effects such as thermal broadening at room temperature may also contribute. Hence, the larger widths used are to assist in replicating the data more closely, with estimates of the widths being determined from fitting Voigt functions to the experimental data. Additionally, the Lorentzian component is given energy dependence to account for the energy dependence of the lifetime[119]. This was done by summing the set width with a factor that varies linearly with energy as implemented in Optados[116].

For the EC and DMC molecules, isolated molecular spectra were generated and energy alignment performed before the spectra were summed. Energy referencing was performed following the method reported in ref. 120, where spectra are initially shifted so the first absorption peak is at 0 eV, and then corrected for the transition energy, calculated as the difference in energy between the ground and excited states. Subsequently, the theoretical curves were rigidly shifted to the data so that the first absorption peaks align to give a better comparison. In cases where the system under investigation possessed more than one inequivalent excitation site, separate core-hole pseudo-potentials and spectra were generated and combined.

## Data availability
The authors declare that the main data supporting the results in this study are available within the paper and its Supplementary Information. The data are available from the corresponding author upon reasonable request.

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

## Acknowledgements

The authors acknowledge funding from the Faraday Institution (faraday.ac.uk; grant numbers FIRG001, FIRG011, FIRG020) and the European Research Council (ERC) under the European Union's Horizon 2020 research and innovation programme (EXISTAR, grant agreement No. 950598) and under the Marie Sklodowska-Curie Actions (ISOBEL, grant agreement No. 101032281). R.J.N. gratefully acknowledges financial support from the EPSRC, grant EP/L022907/1. We thank the ALBA Synchrotron for beamtime on BL 24 (CIRCE) under proposals 2017022094 and 2018022653, and Diamond Light Source for beamtime on Beamline I10 under proposals MM25647 and MM29213, and on Beamline B07 under proposal SI30816-1.

## Author contributions

Electrodes for ex situ and *operando* measurements were prepared by R.S.W. and N.-J.H.K. *Operando* cells were designed by R.S.W. The experimental data was measured by J.E.N.S., M.W.F., N.-J.H.K., J.F.C., C.G.S., C.M.E.P., E.B., and R.S.W. P.B. (I10), C.E. and V.P.D. (BL 24) maintained the beamline endstations and provided experimental support. Data was analysed by J.E.N.S., and M.W.F. with input from C.P.G. and R.S.W. DFT calculations were performed by J.E.N.S. with input and support from R.J.N. The manuscript was written by J.E.N.S. and R.S.W. with input and discussion from all authors. R.S.W. conceived the project and was responsible for obtaining funding and ongoing planning and direction.

## Competing interests

The authors declare no competing interests.
