## [Peer Review File · Nature Communications]

Reviewer Comments, first round

Reviewer #1 (Remarks to the Author):

The manuscript by Swallow et al. uses operando TEY-XAS and TFY-XAS to investigate the SEI formation on silicon anodes for lithium-ion batteries. Although operando TEY-XAS is not new, it is the first time, this method had been applied for the SEI investigation. I appreciate the hard work that went into this study, but unfortunately, this method is very limited for this specific case. This is mainly because the SEI is insulating (as the authors correctly point out) and therefore no electrons can penetrate this barrier (with the exception of some Auger electrons through tunneling). The result is that the sensitivity of TEY-XAS deteriorates with growing SEI and only limited information is obtained, when the SEI is formed, which is not enough to substantiate the claims on the SEI formation and its layered structure. Due to the greater probing depth compared to the spectra recorded at the oxygen K-edge and the F-K-edge, only the Si-K-edge spectra (shown in the SI and not in the main text) give some insight into the silicon species, but these have already been validated by the authors' electrochemical characterizations (shown in figure 2). In fact, the electrochemistry part is in my opinion the strongest. The comparison with the fluoroethylene carbonate additive to stabilize the SEI and to improve the capacity retention is important data, but again the TEY-XAS data (figure 6) does not provide any further insights on the SEI formation.

The authors use then TFY-XAS as an alternative (bulk-sensitive) technique. Their results and spectra are very similar to a recent study by Schellenberger and coworkers on the same system (Materials Today Advances 14 (2022) 100215). It is puzzling, why this work has not been cited here. The authors conclude from their TFY-XAS data (primarily figure 5) that the organic part of the SEI is made of $-(C=O)O^-$ containing species. This is an unsurprising finding, but at the same time not very specific. The small (localized) changes in the peaks could be assigned to many species formed in the SEI. The authors try to only evaluate single peak positions and compare them with literature findings of various possible species. In my view this is insufficient. One should compare the total oxygen absorption spectrum of each individual species if it could fit into the total spectrum. I'm not suggesting that a full multi-component fit should be done, however a qualitative comparison of each species-spectrum to the total spectrum (also in terms of intensity) will greatly improve the analysis.

Their analysis is further based on the increase and decrease of the O-K edge peaks as a function of applied voltage and then rationalize the spectra evolution from their respective peak ratio change at the O-K edge operando spectra (II, III, IV). This approach can be misleading as it can just as easily be argued that the change in the ratio of the peaks is related to saturation effects due to the voltage applied to the electrochemical cell. This should be addressed and discussed in a paragraph. I am not convinced yet that the peaks marked with 'III' in figure 5 are not affected by saturation effects. It would also be nice to see the measured raw spectra in the SI.

Overall, I believe this work is a worthy endeavor that could be published in a more specialized journal: the operando TEY-measurements on the SEI and the comparison with the FEC additive is original, but the method is not novel, and the lack of convincing and/or new results makes it unsuitable for Nature Communications.

Some minor comments:

-The yellow curves and text in figure 2 are hardly visible on the white background.

-Although the authors confirm that the DFT level used in their study is not state-of-the-art, I am a bit disappointed about the poor matching of the calculated spectra with the experimental data, especially for figure 4a. Smith and coworkers (your reference 62) used a similar approach and got better agreement.

Reviewer #2 (Remarks to the Author):

This manuscript presents an *operando* XAS study of SEI formation on amorphous Si with

and without electrolyte additives. To provide additional information, study of a Ni electrode and computational (DFT) studies were also performed. The main result is that FEC additive significantly raises the voltage at which LiF formation occurs. This allows for the substantial volume changes in Si to occur without compromising the protective SEI. Further electrolyte decomposition/electrode degradation is therefore impeded.

This is a careful and interesting study. The use of computational and experimental methods in combination provides a convincing picture of the formation of LiF and organic compounds as a function of voltage and electrolyte composition. The results confirm some previously suspected facts about FEC addition and further clarify the utility of its inclusion. Furthermore the manuscript is well written and engaging. The methods (as pointed out by the authors) are *operando* XAS study is a tour de force demonstration of the power of this methodology and will surely be adopted by other researchers and used for a variety of electrochemical systems. I highly recommend publication in Nature Communications.

I have one tiny comment that the authors may wish to address. In Figure 6, there are two data sets at 1.0V. Like many readers will, I looked at the Figure before reading the text and spent way too much time trying to figure out what the lower (15 minutes later) line was since it is not labeled. After reading the text, it is clear, but not all readers are referees and may be skimming. It would be helpful to label the second curve something like "1.0V after 15 min). Or alternatively, it could be added to the caption. This would allow a more casual reader to understand this plot, which is crucial to the manuscript.

Reviewer #3 (Remarks to the Author):

See report attached

Reviewer #4 (Remarks to the Author):

Characteristics of SEI layer are always a mystery for the battery community. The lack of understanding of SEI is closely related to the fact of the dynamic nature of SEI layer. Therefore, an ideal way to gain information on SEI is using *operando* approaches. This manuscript describes *operando* XAS probing of SEI layer on Si. The key new information appears to be the sequential formation of inorganic (LiF) and organic $-(C=O)O-$ components, which leads to the layer structured SEI. Further, the team also probed the effect of FEC additive, indicating the rapid healing of SEI defects and the improved cycling performance observed. The methodology as described is of general importance for the field of *operando* study of battery using x-ray absorption. The following points should be considered and clarified.

- 1) Is the Ni thin film pure Ni, or a composite of NiO. If so, how does this affect the deconvolution of O spectra.
- 2) It is apparent that the deposited Si is partially oxidized, therefore, the true structural nature of the Si film needs to be clearly characterized. Is it a Si-SiO_x core-shell morphology, or something else.
- 3) How does lithiation lead to Si film morphological evolution?
- 4) During the lithiation of Si, where is the SEI layer, is it on the film surface or is it penetrated into the film? This essentially relates to question 3).
- 5) It is apparent that the spectra of O is contributed by several sources: Oxygen from SiO_x, oxygen from SEI layer, Oxygen from electrolyte. Therefore, how to distinguish each contribution from the total signal.
- 6) The C-edge is not presented, it would be great to show the C-edge information for consistently supporting what is claimed.
- 7) Si spectra is very important, while the analysis of Si spectra appears to be rather neglected. One of the key questions is if in addition to the formation of Li₁₅Si₄, Si appears to be oxidized as well. This aspect of information is missing.

Overall, this is a piece of interesting work to the electrochemical community, while associated with

the poor spatial resolution of the x-ray based techniques, interpretation of data needs careful deliberation of all possible factors.

Reviewer #1 (Remarks to the Author):

The manuscript by Swallow et al. uses operando TEY-XAS and TFY-XAS to investigate the SEI formation on silicon anodes for lithium-ion batteries. Although operando TEY-XAS is not new, it is the first time, this method had been applied for the SEI investigation.

We thank the reviewer for recognising the novelty in our XAS approach and for understanding its practical difficulty.

I appreciate the hard work that went into this study, but unfortunately, this method is very limited for this specific case. This is mainly because the SEI is insulating (as the authors correctly point out) and therefore no electrons can penetrate this barrier (with the exception of some Auger electrons through tunneling). The result is that the sensitivity of TEY-XAS deteriorates with growing SEI and only limited information is obtained, when the SEI is formed, which is not enough to substantiate the claims on the SEI formation and its layered structure.

As the reviewer notes, we explain the interface-sensitivity of TEY-XAS and carefully consider how changes in the measured spectra are influenced by the insulating nature of the SEI layer formed. However, we do not agree with the reviewer's claim that this means only limited information is obtained, and we note that all other reviewers confirm that our TEY-XAS approach yields new insights.

It is important to note that the SEI species observed are clearly visible in the O and F K-edges throughout cycling all the way down to the lowest potential measured (0.1V) – see Figure 3 and Figure 6. Thus the sensitivity of TEY-XAS is clearly sufficient to follow the evolution of these SEI species. This then yields several key insights:

- The potentials at which different species (LiF and $-(C=O)O^-$) form are directly resolved.
- LiF is observed to form prior to the organic $-(C=O)O^-$ species.
- The changes in formation potential due to the presence of the electrolyte additive FEC are directly observed.
- Only when the $-(C=O)O^-$ species appears is the electrolyte signal fully attenuated, indicating the point at which electrical isolation of the electrode from the electrolyte by the SEI is achieved.

We believe these claims regarding the SEI formation are well substantiated by the operando TEY-XAS data presented, and have not been previously verified by an approach that directly probes the evolution of chemical species that make up the SEI. The reviewer is correct that we see attenuation of the electrolyte species as the SEI grows and electrically isolates the electrode from the electrolyte, but in many ways this is beneficial as it avoids numerous overlapping species which could confuse interpretation of the O K-edge signal related to the SEI.

Regarding the SEI having a layered structure, we wish to emphasize that we do not claim to directly observe a layered structure with the data reported here. However, we have observed such layering in our own prior work, and other literature has also reported this which we reference. Herein the TEY-XAS shows the sequential formation of electrolyte components and that the LiF signal becomes more noisy as the $-(C=O)O^-$ species appears. We are careful not to overclaim on this point, and have again checked this and do not believe that at any point we claim that the TEY-XAS data alone verifies this layered structure. For the avoidance of doubt, we include below some of the key sentences from the manuscript related to this point:

“The spectral evolution observed is consistent with the inner SEI (closest to the a-Si) being rich in LiF, while the outer SEI contains more organic species.[26, 30–34] It further indicates that this layering exists from the first SEI formation cycle, and is the result of LiF deposition at

higher potentials followed by organic components as the potential is lowered further, rather than emerging only as a result of repeated decomposition and reduction reactions during ageing.”

“The spectral evolution observed is consistent with layering of the SEI during formation, with LiF deposition occurring close to the electrode surface at higher potentials, followed by organic components forming on top as the potential is further reduced.”

We also highlight in the manuscript that we take a combined approach with FY-XAS:

“Our study of a-Si electrodes herein shows the benefits of combining TEY and FY detection modes under operando conditions, with interface-sensitive TEY being well suited to probing the early stages of SEI formation, whilst FY allows simultaneous probing of the SEI and electrolyte throughout formation.”

Combining the surface sensitivity of TEY and more bulk sensitivity of FY-XAS provides a great deal of information and can be applied to a number of other systems, and we have further work underway in this direction. We therefore respectfully disagree that this methodology only provides limited information, and our data here demonstrates that it can provide interface-sensitive chemical information under electrochemical conditions. As with any technique, there are inherent limitations based on the detection method employed, but useful information is nevertheless extracted.

Due to the greater probing depth compared to the spectra recorded at the oxygen K-edge and the F-K-edge, only the Si-K-edge spectra (shown in the SI and not in the maintext) give some insight into the silicon species, but these have already been validated by the authors' electrochemical characterizations (shown in figure 2). In fact, the electrochemistry part is in my opinion the strongest. The comparison with the fluoroethylene carbonate additive to stabilize the SEI and to improve the capacity retention is important data, but again the TEY-XAS data (figure 6) does not provide any further insights on the SEI formation.

We appreciate the reviewer's acknowledgement of the importance of the electrochemical characterization, and for noting its consistency with the XAS of the Si K-edge. They are correct that the information from the Si K-edge corroborates the cycling data which we view as a strength of our study, confirming that the behaviour observed with our operando methodology is electrochemically representative. However as in our response to the previous comment, we disagree with the statement regarding TEY-XAS not providing any new information. The reviewer has focussed here on the changes in the anode material itself, but the scope of the paper including its title is very much focussed on SEI formation through electrolyte decomposition, where the O and F-edges provide critical insights. Despite the ~10 nm interface-sensitivity when measuring these edges, SEI species are observed throughout our experiments and so it is not the case that only the greater probing depth of the Si K-edge can give insight into the SEI formation.

An example where information can be extracted from TEY-XAS that is not present in the cycling data, is in the importance of the FEC on SEI formation in the data presented in figure 6, where it is highlighted in the text that *“The reduction of FEC at higher potentials than EC has been suggested previously based on electrochemical data[24, 119, 120], but to our knowledge this is the first experimental verification of this using operando spectroscopy, allowing us to identify the potentials where different chemical changes occur.”* The ability to pinpoint the potentials of reduction of the F-products is challenging in electrochemical cycling data but is far clearer in the spectral evolution. We therefore believe Figure 6 gives a very clear insight: FEC reduces at higher potentials than EC leading to LiF formation at higher potentials than without FEC present.

The authors use then TFY-XAS as an alternative (bulk-sensitive) technique. Their results and spectra are very similar to a recent study by Schellenberger and coworkers on the same system (Materials Today Advances 14 (2022) 100215). It is puzzling, why this work has not been cited here.

We appreciate the reviewer pointing out this study, which we became aware of after the submission of our manuscript. The work by Schellenberger and co-workers was published on the 12th March 2022, whilst our study was uploaded to chemrxiv on the 9th March, clarifying why this study wasn't cited in our submission. The most relevant part of this study to ours as pointed out by the reviewer, is in the comparison and similarity of the O K-edge spectra where we add: *"These resemble the main features of the O K-edge XAS spectra reported for LiBF₄ in propylene carbonate (PC), whose oxygen environments are similar to those of EC,[62] and for LiPF₆ in both DMC and EC/DMC[63], as well as X-ray Raman measurements of LiPF₆ in both PC and EC/DMC.[64] We therefore attribute these features to the electrolyte solvents EC/DMC."*

Schellenberger's work is based on transmission XAS (rather than the TEY-XAS or TFY-XAS used herein) and involves X-ray beam induced electrolyte decomposition at a certain point in cycling, so that a bubble is formed to allow soft X-ray transmission. Our *operando* approach is distinct in that the cell environment remains intact throughout cycling, and continuous measurements can be performed during electrochemical cycling.

The authors conclude from their TFY-XAS data (primarily figure 5) that the organic part of the SEI is made of $-(C=O)O-$ containing species. This is an unsurprising finding, but at the same time not very specific.

We disagree that our assignment of $-(C=O)O-$ containing species is unspecific. We identify a specific molecular motif within the limitations that soft XAS provides information on local electronic structure. We further discuss the chemical routes by which these species can form based on the electrolyte species present and prior literature. Although LEDC/LEMC are more commonly assigned organic SEI components, we reference prior literature where $-(C=O)O-$ containing species are assigned, noting that these provide limited information on when during SEI formation these species appear. As the reviewer points out further below, there are challenges in deconvoluting XAS spectra where multiple species can potential overlap, and we have taken care within these limitations to rationalise and specify the dominant chemical species to an appropriate level, whilst still providing useful and interesting insights into the mechanisms of SEI formation.

The small (localized) changes in the peaks could be assigned to many species formed in the SEI. The authors try to only evaluate single peak positions and compare them with literature findings of various possible species. In my view this is insufficient. One should compare the total oxygen absorption spectrum of each individual species if it could fit into the total spectrum. I'm not suggesting that a full multi-component fit should be done, however a qualitative comparison of each species-spectrum to the total spectrum (also in terms of intensity) will greatly improve the analysis.

We appreciate the reviewer highlighting this potential issue, however we can reassure them that we do not only evaluate single peak positions from literature. In fact, we look at changes across the whole spectrum and make qualitative assignments based on sound reasoning regarding the electrolyte environment and the decomposition reactions that can occur. An example of this is in the analysis of the peaks in figure 5 that the reviewer discusses in their next point. As they note, we consider the peak ratios changes of peaks II, III, and IV highlighting that it is not only single peak positions being considered but rather the total spectrum.

In Figure 5b, we do compare the peak positions for a variety of species with peak I, so we apologise if this gave the incorrect impression that only the peak I* position was considered in assigning the organic SEI species. As already mentioned, we do compare with the total spectrum in the main text. Therefore to make this more clear, we have now included reference spectra in figure 5a-b for materials which share the structural motif -C(=O)O- in the form of Li acetate, Li oxalate and Li formate (and added associated text). These full spectra are plotted to allow direct comparison with the operando FY-XAS, and further support the arguments previously made in the text about species assignment i.e. -C(=O)- motifs such as aldehyde and ketone groups contribute little to the spectral intensity above ~ 535 eV so cannot account for the change in peak III intensity, whilst -C(=O)O- motifs do contribute intensity in the correct energy range. We believe this is the appropriate level of comparison for the assignment made, and agree with the reviewer that going further (such as a multi-component fit of the data) is likely to only complicate the analysis, raising questions such as which reference spectra should be included in the fit.

Their analysis is further based on the increase and decrease of the O-K edge peaks as a function of applied voltage and then rationalize the spectra evolution from their respective peak ratio change at the O-K edge operando spectra (II, III, IV). This approach can be misleading as it can just as easily be argued that the change in the ratio of the peaks is related to saturation effects due to the voltage applied to the electrochemical cell. This should be addressed and discussed in a paragraph. I am not convinced yet that the peaks marked with 'III' in figure 5 are not affected by saturation effects. It would also be nice to see the measured raw spectra in the SI.

We thank the reviewer for their careful consideration of the FY data displayed in figure 5, and highlighting the potential pitfalls related to saturations effects. We fully agree that due care is required in analysing FY data to avoid incorrectly attributing saturation effects to chemical changes. To address the specific concern about whether the changes in peak ratios are a voltage-dependent saturation effect (we assume the reviewer is suggesting this might arise due to the rearrangement of ions in the electrolyte under bias), we have now added a figure in the SI showing negligible change in the O K-edge FY spectra for a cell held at 0.05V before and after removal of the bias. This can be easily rationalised by considering that the Debye length for the 1M solution used herein is ~ 2 nm (calculated using Debye-Hückel formalism for an electrolyte solution). This is two orders of magnitude smaller than the X-ray attenuation length of several hundred nm for such organic electrolytes, and thus the lack of noticeable change related to saturation effects is to be expected.

We have added the following text in the main manuscript to address this point: "We demonstrate that the changes seen do not simply arise from changes in self-absorption/saturation effects due to ion rearrangement under applied potential, as when the bias is removed from the cell the spectral shape remains unchanged (see supplementary Figure S10)"

The effects of self-absorption in FY-XAS are a common problem, modifying the spectral shape such that the highest peaks appear compressed with respect to the lower peaks (see: de Groot, Kotani et al. *Core Level Spectroscopy of Solids*, CRC press). However, here we are seeing peaks of similar intensity, e.g. peak I and III, where one grows whilst the other shrinks. Meanwhile nearby features such as peak II (higher intensity) and peak IV (lower intensity) show negligible relative intensity change. This is not consistent with a self-absorption effect where the more intense peaks would all either shrink or grow together with respect to the less intense ones, depending on whether self-absorption was becoming more or less severe as the SEI forms. On this basis, although self-absorption may be affecting peaks in the spectrum to some extent, the growth in intensity of peak III alongside peak I can't be well explained by self-absorption/saturation effects, and thus we attribute these to new chemical species being formed through electrolyte decomposition as outlined in the original manuscript. Therefore,

following the reviewer's suggestion we also add the following paragraph to the main manuscript:

"Note that changes in spectral shape due to the geometrical effects of self-absorption and saturation related to measurements in FY mode[111–113] are complex in this system due to the measured signal coming from multiple layers of different thicknesses and densities, with one growing electrochemically during the measurement. This of course makes any correction schemes, which typically make simplifying assumptions regarding the sample geometry, very difficult to implement.[114-117] However, whilst we do not claim this data is free from self-absorption or saturation effects, assuming smoothly varying absorption coefficients (apart from at the step edge)[118], and noting that the measurement geometry stays fixed throughout the experiment and that the SEI layer is much thinner than the X-ray attenuation length, we can expect a uniform effect across the spectra, i.e. contiguous features of similar intensity should not both grow and shrink across the energy range. Hence, whilst the FY spectra can't be assumed to accurately map the absorption coefficient across the O-edge, the changes seen are consistent with chemical changes rather than geometric ones."

The reviewer mentions it would be nice to see the measured raw spectra in the SI. We assume that the reviewer is interested in comparing between self-absorption corrected and uncorrected spectra. We wish to emphasise however that no correction for self-absorption has been made in any of the data presented (as noted in the text added above). The spectra as shown in the manuscript are essentially the raw spectra, with the collected data having been divided by the I_0 signal, had a linear background subtracted, and been normalised for graphical comparison.

Overall, I believe this work is a worthy endeavor that could be published in a more specialized journal: the operando TEY-measurements on the SEI and the comparison with the FEC additive is original, but the method is not novel, and the lack of convincing and/or new results makes it unsuitable for Nature Communications.

We appreciate the reviewer's comments and hope the changes made and clarification provided based on these should help convince them of the work's novelty, credibility and suitability for Nature Communications.

Some minor comments:

-The yellow curves and text in figure 2 are hardly visible on the white background.

Thank you for pointing this out, we have made the yellow curve and text in figure 2 darker so it is more visible.

-Although the authors confirm that the DFT level used in their study is not state-of-the-art, I am a bit disappointed about the poor matching of the calculated spectra with the experimental data, especially for figure 4a. Smith and coworkers (your reference 62) used a similar approach and got better agreement

As stated in the text, the DFT used in this study utilises the appropriate level of computation to provide an adequate description of the spectra, in this sense it is still "state of the art". In the paper by Smith et al. they apply molecular dynamics (which uses the theory of classical force fields and is thus not DFT in itself), to obtain a starting geometry of their similar molecular system. They go on to compare the influence of different concentrations of lithium salts in their electrolyte, motivating their molecular dynamic approach. In our manuscript, DFT is used to assign the origins of spectral features which it does adequately, and thus the increased computational cost associated with the large cells generated by MD is not well justified. As we explain in the manuscript, the MD approach would yield broader features matching experiment better, but is not expected to provide additional insight to this study.

Reviewer #2 (Remarks to the Author):

This manuscript presents an operando XAS study of SEI formation on amorphous Si with and without electrolyte additives. To provide additional information, study of a Ni electrode and computational (DFT) studies were also performed. The main result is that FEC additive significantly raises the voltage at which LiF formation occurs. This allows for the substantial volume changes in Si to occur without compromising the protective SEI. Further electrolyte decomposition/electrode degradation is therefore impeded.

This is a careful and interesting study. The use of computational and experimental methods in combination provides a convincing picture of the formation of LiF and organic compounds as a function of voltage and electrolyte composition. The results confirm some previously suspected facts about FEC addition and further clarify the utility of its inclusion. Furthermore the manuscript is well written and engaging. The methods (as pointed out by the authors) are operando XAS study is a tour de force demonstration of the power of this methodology and will surely be adopted by other researchers and used for a variety of electrochemical systems. I highly recommend publication in Nature Communications.

We very much appreciate the reviewer's positive appraisal of our work, and succinctly highlighting the areas where it provides new insights.

I have one tiny comment that the authors may wish to address. In Figure 6, there are two data sets at 1.0V. Like many readers will, I looked at the Figure before reading the text and spent way too much time trying to figure out what the lower (15 minutes later) line was since it is not labeled. After reading the text, it is clear, but not all readers are referees and may be skimming. It would be helpful to label the second curve something like "1.0V after 15 min). Or alternatively, it could be added to the caption. This would allow a more casual reader to understand this plot, which is crucial to the manuscript.

We appreciate the suggestion which is intended to more clearly convey the information within the paper to potential readers. Therefore, we now include a figure label highlighting that the 2nd spectrum at 1.0 V was recorded 15 mins after the previous one. We also include an adjustment to the figure caption to that affect.

Reviewer #3 (Remarks to the Author):

This paper is a very nice piece of Science. It represents a very important advancement in the field of operando spectroscopy, and I do not see any serious reason preventing its publication in Nature Communications. I just have one single question for the Authors. They developed a nice experimental method for separating the current coming from the battery from the photocurrent due to X-ray illumination. Usually this last current is in the range of nanoampere or less, while the current from the battery is many orders of magnitude larger. Could the Authors further comment on this point? I will be quite curious to understand the details of their experiments in this respect.

We appreciate the reviewer's positive assessment of our manuscript, and their confirmation that this is an important advancement in operando methodology. To answer the reviewer's question, typically the current measured from the TEY signal was <5 nA whilst the current under electrochemical cycling was ~250 nA, around two orders of magnitude greater. During voltage holds used for each measurement, the faradaic will vary throughout the measurement. Lock-in based approaches are well suited for extracting small signals from a larger noisy or slowly varying signal and are found to be effective in this application. We now include a statement to this effect when discussing the experimental set-up of the XAS: "A SR830 lock-in amplifier (Stanford Research Systems) is then used to separate the modulated TEY current (<5 nA) from the faradaic current (~250 nA)."

Reviewer #4 (Remarks to the Author):

Characteristics of SEI layer are always a mystery for the battery community. The lack of understanding of SEI is closely related to the fact of the dynamic nature of SEI layer. Therefore, an ideal way to gain information on SEI is using operando approaches. This manuscript describes operando XAS probing of SEI layer on Si. The key new information appears to be the sequential formation of inorganic (LiF) and organic (-(C=O)O-) components, which leads to the layer structured SEI. Further, the team also probed the effect of FEC additive, indicating the rapid healing of SEI defects and the improved cycling performance observed. The methodology as described is of general importance for the field of operando study of battery using x-ray absorption. The following points should be considered and clarified.

We thank the reviewer for confirming the general importance of the methodology described and several of the new insights about the SEI that it has been able to provide. We address their more specific points below.

1) Is the Ni thin film pure Ni, or a composite of NiO. If so, how does this affect the deconvolution of O spectra.

The Ni thin film is sputter deposited in a vacuum chamber with Ar as the sputtering gas, leading to the deposition of a metallic Ni film. However, exposure to air on removal from the deposition chamber leads to formation of a surface oxide that is indeed apparent in the O K-edge spectrum as can be seen in figure S5. However, figure S5 also confirms that when 20 nm of a-Si is sputter deposited on top of this layer the feature related to NiO no longer contributes to the O K-edge.

To make this clearer, we have slightly adjusted the text in the SI to more clearly explain why NiO does not contribute to the O K-edge spectra following a-Si deposition:

“This NiO peak is no longer seen when a-Si(20 nm) is deposited on top, consistent with the NiO being buried at a depth greater than the ~10 nm range of electrons detected by TEY-XAS of the O K-edge. It may also be the case that the thin NiO layer is sputtered away to some extent by the energetic Si atoms impinging during sputter-deposition of the a-Si layer.”

We further note that for uncovered Ni films, ongoing TEY-XAS studies for the same electrolyte system have revealed that Ni is fully reduced by 2 V vs. Li. We intend to publish these results in a separate study and believe they fall beyond the scope of this paper. However, this gives confidence that significant NiO contributions are not expected, even for the FY-XAS of Ni films shown in Figure 5.

2) It is apparent that the deposited Si is partially oxidized, therefore, the true structural nature of the Si film needs to be clearly characterized. Is it a Si-SiO_x core-shell morphology, or something else.

The thin film electrodes are sputter deposited as elemental Si, and so are not expected to have a core-shell morphology which would more typically be associated with spherical particles. However, much like the Ni our Si thin films have a native oxide layer due to air exposure following deposition as shown by the TEY-XAS data in Figure S5. The electrochemical cycling displayed in Figure 2 gives confidence that the silicon thin film is amorphous as there are no peaks related to crystalline Si restructuring. Nevertheless, we performed additional Raman spectroscopy of the as-deposited film to further confirm this which is now included in the supporting information (Figure S1, with associated text in the main manuscript: “The amorphous phase of the Si film was confirmed by Raman spectroscopy (see Figure S1), using a Reinshaw inVia Raman microscope with backscattering geometry. A laser wavelength of 532 nm was used, with a spot diameter of 1-2 μm and an operating power of 0.2 mW focused through an inverted microscope via a 50× objective lens.”

3) How does lithiation lead to Si film morphological evolution?

We have now provided SEM data in Fig. 2 (and associated text) to address this point. To briefly summarise, during the first full cycle where the SEI first forms, very little change in electrode morphology is observed. However, more extended cycling (30 cycles) leads to cracking of the Si surface where additional SEI formation is expected. This doesn't affect the conclusions of the paper, where the operando data concerns the first half cycle where significant cracking is not observed.

4) During the lithiation of Si, where is the SEI layer, is it on the film surface or is it penetrated into the film? This essentially relates to question 3).

During the formation of the SEI studied herein, SEM data (now included in Figure 2) indicates the a-Si remains as a continuous film during the first half cycle, and thus the SEI layer forms on the surface of this film which is in contact with the electrolyte. Similar to our response for point 3), we have provided a description of the SEM data that is now included, which addresses this question:

"Figure 2d-g shows SEM of a-Si cells cycled in LP30 electrolyte (no additive) and stopped at different stages of cycling revealing changes in the electrode morphology. Initially, the pristine a-Si electrode (Figure 2d)) appears relatively smooth, showing topography that matches the rolling striations of the underlying Cu substrate. On cycling to 5 mV during the 1st cycle (Figure 2e), the electrode surface remains similarly smooth despite the large expected volume increase due to lithiation of the a-Si, indicating this is primarily accommodated through swelling of the electrode thickness. On cycling back to 2 V (Figure 2f), the overall surface morphology remains largely unchanged although a small amount of cracking can be discerned close to distinct topographic features such as striations. Therefore, during the first cycle the thin film electrode remains continuous with SEI formation expected to occur predominantly at the exposed electrode surface rather than penetrating through the electrode thickness. Close inspection of Figure 2e reveals several small bright dots $<<1$ μm in lateral dimensions, however these are not as apparent in Figure 2f. These are likely products formed during cell disassembly and inert transfer, reflecting the high reactivity of the lithiated a-Si. Following more extended cycling (30 cycles, Figure 2g) much larger morphological changes are apparent with an interconnected network of cracks apparent at 2 V. This is attributable to contraction of the lithiated a-Si as it is delithiated at high potentials leading to the formation of silicon islands of <1 μm in at least one direction, which are separated by sizable gaps of ~ 200 nm. These provide pathways for electrolyte to penetrate and form fresh SEI through the electrode thickness. These cracks are seen to be refilled to some extent through expansion of the a-Si when it is again lithiated (inset of Figure 2g). This repeated cracking and SEI formation eventually leads to isolation and/or delamination of Si islands from the current collector, contributing to the capacity fade observed after repeated cycling."

5) It is apparent that the spectra of O is contributed by several sources: Oxygen from SiO_x, oxygen from SEI layer, Oxygen from electrolyte. Therefore, how to distinguish each contribution from the total signal.

As noted in the response to reviewer 1, the overlap of spectral features from different species is a common challenge for experimental spectroscopists, particularly when studying more realistic and complex chemical environments. We have addressed several of the points raised here in our response to reviewer 1, but provide a further summary here explaining how the different oxygen species are distinguished.

Our assignments of spectral features are based on both reference spectra (both from this work and prior literature), and spectral simulations based on DFT calculations. The spectra of SiO_x and other related references are included in the supplementary information. Simulated spectra of the electrolyte species are presented in the main manuscript and correspond well with prior literature references that are discussed. For the SEI components we have now added additional reference spectra for candidate SEI species. Whilst there are regions of the spectra where large amount of spectral overlap between these different species does occur, we base our analysis on regions where there is minimal spectral overlap and/or where there are characteristic peaks that aren't present in the other species. For example, peak I^* is low in energy, and compared to the SiO_x references, the one containing NiO is the only possible overlapping component. However, as explained in our response to point 1, the Ni electrode always becomes reduced upon cycling, thus cannot contribute to this peak at low potentials.

6) The C-edge is not presented, it would be great to show the C-edge information for consistently supporting what is claimed.

We appreciate the reviewer's comment and agree that studies of the C K-edge have the potential to provide further insight into SEI formation in organic electrolytes. However, such studies are made challenging by the prevalence (and accumulation with time) of carbonaceous contamination on many surfaces. This leads to dips in X-ray intensity at the C K-edge for many beamlines due to carbon contamination on optical components such as mirrors. This can completely distort the measured C K-edge signal and can't always be fully removed by I_0 correction. Indeed, for our operando cells it is not immediately obvious how a suitable I_0 measurement could be performed that takes account of adventitious carbon on the outer surface of the silicon nitride windows used to seal them. Additionally, the 100 nm thick silicon nitride windows are much less transparent to incident photons at the energies needed for acquiring the C K-edge compared to the O or F K-edges. Addressing this requires further development of windows which are thinner and/or made from different materials but are still sufficiently strong to maintain the pressure difference between the liquid environment inside the cell and the surrounding vacuum. We hope that this can be achieved in the future, but hopefully this satisfies the reviewer as to why C K-edges are not be included in the present study.

7) Si spectra is very important, while the analysis of Si spectra appears to be rather neglected. One of the key questions is if in addition to the formation of $\text{Li}_{15}\text{Si}_4$, Si appears to be oxidized as well. This aspect of information is missing.

As we noted in our earlier response to reviewer 1, and made clear in the manuscript, the main focus of our study is the SEI formation, rather than the lithiation of Si. We agree however, that there is important information to be obtained from the Si spectra, and we include operando Si K-edge data in the supplementary information (Figure S4) for exactly this reason. We confirm that we have carefully considered and analysed this data, and fully agree with the reviewer that whether the Si remains oxidised at low potentials is a key question. Our data reveals that the Si is reduced by 0.6 V prior to significant lithiation, as seen by the absence of the SiO_2 feature in the Si K-edge compared to the as-deposited a-Si (i.e. air transferred). This data also confirms that lithiation of the Si has occurred by 0.2 V. We directly address this point in the main manuscript:

"Although the focus herein is the SEI components, Si K-edge measurements were also performed at several potentials confirming removal of SiO_2 from the Si surface and the lithiation of Si at low potentials (see supplementary Figure S4), as expected from the cycling data presented in Figure 2."

As pointed out by reviewer 1, we also provide a detailed account of the silicon lithiation based on the electrochemical data of Figure 2, where signatures for the different Si lithiation

processes are much more well-established and apparent, in contrast to the SEI formation processes. We thus believe the information requested by the reviewer regarding the Si spectra and chemical state of the Si is included in the manuscript. To address the reviewer's comment we have slightly edited the text in the SI accompanying Figure S4 to highlight the SiO₂ reduction.

Overall, this is a piece of interesting work to the electrochemical community, while associated with the poor spatial resolution of the x-ray based techniques, interpretation of data needs careful deliberation of all possible factors

We thank the reviewer for their positive assessment of our work and the interesting questions they pose which we have addressed above. Whilst we agree that the lateral resolution of our soft XAS approach is limited due to the relatively large X-ray spot size, the interface sensitivity of ~10 nm and ability to resolve different chemical species provide important insights into SEI formation processes. Within this context we have been careful to interpret our data within the limitations of the techniques applied such that robust conclusions are drawn.

Reviewer Comments, second round

Reviewer #3 (Remarks to the Author):

The Authors have replied to my previous comment, and I now can recommend this paper for publication.

Reviewer #4 (Remarks to the Author):

Upon revision, the authors have addressed my question very carefully. The revised manuscript is in good standing and I get no objection for publication in the present form.

Original Reviewer Comments

Reviewer #1 (Remarks to the Author):

The manuscript by Swallow et al. uses operando TEY-XAS and TFY-XAS to investigate the SEI formation on silicon anodes for lithium-ion batteries. Although operando TEY-XAS is not new, it is the first time, this method had been applied for the SEI investigation.

We thank the reviewer for recognising the novelty in our XAS approach and for understanding its practical difficulty.

I appreciate the hard work that went into this study, but unfortunately, this method is very limited for this specific case. This is mainly because the SEI is insulating (as the authors correctly point out) and therefore no electrons can penetrate this barrier (with the exception of some Auger electrons through tunneling). The result is that the sensitivity of TEY-XAS deteriorates with growing SEI and only limited information is obtained, when the SEI is formed, which is not enough to substantiate the claims on the SEI formation and its layered structure.

As the reviewer notes, we explain the interface-sensitivity of TEY-XAS and carefully consider how changes in the measured spectra are influenced by the insulating nature of the SEI layer formed. However, we do not agree with the reviewer's claim that this means only limited information is obtained, and we note that all other reviewers confirm that our TEY-XAS approach yields new insights.

It is important to note that the SEI species observed are clearly visible in the O and F K-edges throughout cycling all the way down to the lowest potential measured (0.1V) – see Figure 3 and Figure 6. Thus the sensitivity of TEY-XAS is clearly sufficient to follow the evolution of these SEI species. This then yields several key insights:

- The potentials at which different species (LiF and $-(C=O)O-$) form are directly resolved.
- LiF is observed to form prior to the organic $-(C=O)O-$ species.
- The changes in formation potential due to the presence of the electrolyte additive FEC are directly observed.
- Only when the $-(C=O)O-$ species appears is the electrolyte signal fully attenuated, indicating the point at which electrical isolation of the electrode from the electrolyte by the SEI is achieved.

We believe these claims regarding the SEI formation are well substantiated by the operando TEY-XAS data presented, and have not been previously verified by an approach that directly probes the evolution of chemical species that make up the SEI. The reviewer is correct that we see attenuation of the electrolyte species as the SEI grows and electrically isolates the electrode from the electrolyte, but in many ways this is beneficial as it avoids numerous overlapping species which could confuse interpretation of the O K-edge signal related to the SEI.

Regarding the SEI having a layered structure, we wish to emphasize that we do not claim to directly observe a layered structure with the data reported here. However, we have observed such layering in our own prior work, and other literature has also reported this which we reference. Herein the TEY-XAS shows the sequential formation of electrolyte components and that the LiF signal becomes more noisy as the $-(C=O)O-$ species appears. We are careful not to overclaim on this point, and have again checked this and do not believe that at any point we claim that the TEY-XAS data alone verifies this layered structure. For the avoidance of doubt, we include below some of the key sentences from the manuscript related to this point:

“The spectral evolution observed is consistent with the inner SEI (closest to the a-Si) being rich in LiF, while the outer SEI contains more organic species.[26, 30–34] It further indicates that this layering exists from the first SEI formation cycle, and is the result of LiF deposition at higher potentials followed by organic components as the potential is lowered further, rather than emerging only as a result of repeated decomposition and reduction reactions during ageing.”

“The spectral evolution observed is consistent with layering of the SEI during formation, with LiF deposition occurring close to the electrode surface at higher potentials, followed by organic components forming on top as the potential is further reduced.”

We also highlight in the manuscript that we take a combined approach with FY-XAS:

“Our study of a-Si electrodes herein shows the benefits of combining TEY and FY detection modes under operando conditions, with interface-sensitive TEY being well suited to probing the early stages of SEI formation, whilst FY allows simultaneous probing of the SEI and electrolyte throughout formation.”

Combining the surface sensitivity of TEY and more bulk sensitivity of FY-XAS provides a great deal of information and can be applied to a number of other systems, and we have further work underway in this direction. We therefore respectfully disagree that this methodology only provides limited information, and our data here demonstrates that it can provide interface-sensitive chemical information under electrochemical conditions. As with any technique, there are inherent limitations based on the detection method employed, but useful information is nevertheless extracted.

Due to the greater probing depth compared to the spectra recorded at the oxygen K-edge and the F-K-edge, only the Si-K-edge spectra (shown in the SI and not in the maintext) give some insight into the silicon species, but these have already been validated by the authors' electrochemical characterizations (shown in figure 2). In fact, the electrochemistry part is in my opinion the strongest. The comparison with the fluoroethylene carbonate additive to stabilize the SEI and to improve the capacity retention is important data, but again the TEY-XAS data (figure 6) does not provide any further insights on the SEI formation.

We appreciate the reviewer's acknowledgement of the importance of the electrochemical characterization, and for noting its consistency with the XAS of the Si K-edge. They are correct that the information from the Si K-edge corroborates the cycling data which we view as a strength of our study, confirming that the behaviour observed with our operando methodology is electrochemically representative. However as in our response to the previous comment, we disagree with the statement regarding TEY-XAS not providing any new information. The reviewer has focussed here on the changes in the anode material itself, but the scope of the paper including its title is very much focussed on SEI formation through electrolyte decomposition, where the O and F-edges provide critical insights. Despite the ~10 nm interface-sensitivity when measuring these edges, SEI species are observed throughout our experiments and so it is not the case that only the greater probing depth of the Si K-edge can give insight into the SEI formation.

An example where information can be extracted from TEY-XAS that is not present in the cycling data, is in the importance of the FEC on SEI formation in the data presented in figure 6, where it is highlighted in the text that *“The reduction of FEC at higher potentials than EC has been suggested previously based on electrochemical data[24, 119, 120], but to our knowledge this is the first experimental verification of this using operando spectroscopy, allowing us to identify the potentials where different chemical changes occur.”* The ability to pinpoint the potentials of reduction of the F-products is challenging in electrochemical cycling

data but is far clearer in the spectral evolution. We therefore believe Figure 6 gives a very clear insight: FEC reduces at higher potentials than EC leading to LiF formation at higher potentials than without FEC present.

The authors use then TFY-XAS as an alternative (bulk-sensitive) technique. Their results and spectra are very similar to a recent study by Schellenberger and coworkers on the same system (Materials Today Advances 14 (2022) 100215). It is puzzling, why this work has not been cited here.

We appreciate the reviewer pointing out this study, which we became aware of after the submission of our manuscript. The work by Schellenberger and co-workers was published on the 12th March 2022, whilst our study was uploaded to chemrxiv on the 9th March, clarifying why this study wasn't cited in our submission. The most relevant part of this study to ours as pointed out by the reviewer, is in the comparison and similarity of the O K-edge spectra where we add: *“These resemble the main features of the O K-edge XAS spectra reported for LiBF₄ in propylene carbonate (PC), whose oxygen environments are similar to those of EC,[62] and for LiPF₆ in both DMC and EC/DMC[63], as well as X-ray Raman measurements of LiPF₆ in both PC and EC/DMC.[64] We therefore attribute these features to the electrolyte solvents EC/DMC.”*

Schellenberger's work is based on transmission XAS (rather than the TEY-XAS or TFY-XAS used herein) and involves X-ray beam induced electrolyte decomposition at a certain point in cycling, so that a bubble is formed to allow soft X-ray transmission. Our *operando* approach is distinct in that the cell environment remains intact throughout cycling, and continuous measurements can be performed during electrochemical cycling.

The authors conclude from their TFY-XAS data (primarily figure 5) that the organic part of the SEI is made of $-(C=O)O-$ containing species. This is an unsurprising finding, but at the same time not very specific.

We disagree that our assignment of $-(C=O)O-$ containing species is unspecific. We identify a specific molecular motif within the limitations that soft XAS provides information on local electronic structure. We further discuss the chemical routes by which these species can form based on the electrolyte species present and prior literature. Although LEDC/LEMC are more commonly assigned organic SEI components, we reference prior literature where $-(C=O)O-$ containing species are assigned, noting that these provide limited information on when during SEI formation these species appear. As the reviewer points out further below, there are challenges in deconvoluting XAS spectra where multiple species can potential overlap, and we have taken care within these limitations to rationalise and specify the dominant chemical species to an appropriate level, whilst still providing useful and interesting insights into the mechanisms of SEI formation.

The small (localized) changes in the peaks could be assigned to many species formed in the SEI. The authors try to only evaluate single peak positions and compare them with literature findings of various possible species. In my view this is insufficient. One should compare the total oxygen absorption spectrum of each individual species if it could fit into the total spectrum. I'm not suggesting that a full multi-component fit should be done, however a qualitative comparison of each species-spectrum to the total spectrum (also in terms of intensity) will greatly improve the analysis.

We appreciate the reviewer highlighting this potential issue, however we can reassure them that we do not only evaluate single peak positions from literature. In fact, we look at changes across the whole spectrum and make qualitative assignments based on sound reasoning regarding the electrolyte environment and the decomposition reactions that can occur. An example of this is in the analysis of the peaks in figure 5 that the reviewer discusses in their

next point. As they note, we consider the peak ratios changes of peaks II, III, and IV highlighting that it is not only single peak positions being considered but rather the total spectrum.

In Figure 5b, we do compare the peak positions for a variety of species with peak I, so we apologise if this gave the incorrect impression that only the peak I* position was considered in assigning the organic SEI species. As already mentioned, we do compare with the total spectrum in the main text. Therefore to make this more clear, we have now included reference spectra in figure 5a-b for materials which share the structural motif -C(=O)O- in the form of Li acetate, Li oxalate and Li formate (and added associated text). These full spectra are plotted to allow direct comparison with the operando FY-XAS, and further support the arguments previously made in the text about species assignment i.e. -C(=O)- motifs such as aldehyde and ketone groups contribute little to the spectral intensity above ~ 535 eV so cannot account for the change in peak III intensity, whilst -C(=O)O- motifs do contribute intensity in the correct energy range. We believe this is the appropriate level of comparison for the assignment made, and agree with the reviewer that going further (such as a multi-component fit of the data) is likely to only complicate the analysis, raising questions such as which reference spectra should be included in the fit.

Their analysis is further based on the increase and decrease of the O-K edge peaks as a function of applied voltage and then rationalize the spectra evolution from their respective peak ratio change at the O-K edge operando spectra (II, III, IV). This approach can be misleading as it can just as easily be argued that the change in the ratio of the peaks is related to saturation effects due to the voltage applied to the electrochemical cell. This should be addressed and discussed in a paragraph. I am not convinced yet that the peaks marked with 'III' in figure 5 are not affected by saturation effects. It would also be nice to see the measured raw spectra in the SI.

We thank the reviewer for their careful consideration of the FY data displayed in figure 5, and highlighting the potential pitfalls related to saturations effects. We fully agree that due care is required in analysing FY data to avoid incorrectly attributing saturation effects to chemical changes. To address the specific concern about whether the changes in peak ratios are a voltage-dependent saturation effect (we assume the reviewer is suggesting this might arise due to the rearrangement of ions in the electrolyte under bias), we have now added a figure in the SI showing negligible change in the O K-edge FY spectra for a cell held at 0.05V before and after removal of the bias. This can be easily rationalised by considering that the Debye length for the 1M solution used herein is ~ 2 nm (calculated using Debye-Hückel formalism for an electrolyte solution). This is two orders of magnitude smaller than the X-ray attenuation length of several hundred nm for such organic electrolytes, and thus the lack of noticeable change related to saturation effects is to be expected.

We have added the following text in the main manuscript to address this point: "We demonstrate that the changes seen do not simply arise from changes in self-absorption/saturation effects due to ion rearrangement under applied potential, as when the bias is removed from the cell the spectral shape remains unchanged (see supplementary Figure S10)"

The effects of self-absorption in FY-XAS are a common problem, modifying the spectral shape such that the highest peaks appear compressed with respect to the lower peaks (see: de Groot, Kotani et al. *Core Level Spectroscopy of Solids*, CRC press). However, here we are seeing peaks of similar intensity, e.g. peak I and III, where one grows whilst the other shrinks. Meanwhile nearby features such as peak II (higher intensity) and peak IV (lower intensity) show negligible relative intensity change. This is not consistent with a self-absorption effect where the more intense peaks would all either shrink or grow together with respect to the less intense ones, depending on whether self-absorption was becoming more or less severe as the SEI forms. On this basis, although self-absorption may be affecting peaks in the spectrum

to some extent, the growth in intensity of peak III alongside peak I can't be well explained by self-absorption/saturation effects, and thus we attribute these to new chemical species being formed through electrolyte decomposition as outlined in the original manuscript. Therefore, following the reviewer's suggestion we also add the following paragraph to the main manuscript:

"Note that changes in spectral shape due to the geometrical effects of self-absorption and saturation related to measurements in FY mode[111–113] are complex in this system due to the measured signal coming from multiple layers of different thicknesses and densities, with one growing electrochemically during the measurement. This of course makes any correction schemes, which typically make simplifying assumptions regarding the sample geometry, very difficult to implement.[114-117] However, whilst we do not claim this data is free from self-absorption or saturation effects, assuming smoothly varying absorption coefficients (apart from at the step edge)[118], and noting that the measurement geometry stays fixed throughout the experiment and that the SEI layer is much thinner than the X-ray attenuation length, we can expect a uniform effect across the spectra, i.e. contiguous features of similar intensity should not both grow and shrink across the energy range. Hence, whilst the FY spectra can't be assumed to accurately map the absorption coefficient across the O-edge, the changes seen are consistent with chemical changes rather than geometric ones."

The reviewer mentions it would be nice to see the measured raw spectra in the SI. We assume that the reviewer is interested in comparing between self-absorption corrected and uncorrected spectra. We wish to emphasise however that no correction for self-absorption has been made in any of the data presented (as noted in the text added above). The spectra as shown in the manuscript are essentially the raw spectra, with the collected data having been divided by the I_0 signal, had a linear background subtracted, and been normalised for graphical comparison.

Overall, I believe this work is a worthy endeavor that could be published in a more specialized journal: the operando TEY-measurements on the SEI and the comparison with the FEC additive is original, but the method is not novel, and the lack of convincing and/or new results makes it unsuitable for Nature Communications.

We appreciate the reviewer's comments and hope the changes made and clarification provided based on these should help convince them of the work's novelty, credibility and suitability for Nature Communications.

Some minor comments:

-The yellow curves and text in figure 2 are hardly visible on the white background.

Thank you for pointing this out, we have made the yellow curve and text in figure 2 darker so it is more visible.

-Although the authors confirm that the DFT level used in their study is not state-of-the-art, I am a bit disappointed about the poor matching of the calculated spectra with the experimental data, especially for figure 4a. Smith and coworkers (your reference 62) used a similar approach and got better agreement

As stated in the text, the DFT used in this study utilises the appropriate level of computation to provide an adequate description of the spectra, in this sense it is still "state of the art". In the paper by Smith et al. they apply molecular dynamics (which uses the theory of classical force fields and is thus not DFT in itself), to obtain a starting geometry of their similar molecular system. They go on to compare the influence of different concentrations of lithium salts in their electrolyte, motivating their molecular dynamic approach. In our manuscript, DFT is used to assign the origins of spectral features which it does adequately, and thus the increased

computational cost associated with the large cells generated by MD is not well justified. As we explain in the manuscript, the MD approach would yield broader features matching experiment better, but is not expected to provide additional insight to this study.

Reviewer #2 (Remarks to the Author):

This manuscript presents an operando XAS study of SEI formation on amorphous Si with and without electrolyte additives. To provide additional information, study of a Ni electrode and computational (DFT) studies were also performed. The main result is that FEC additive significantly raises the voltage at which LiF formation occurs. This allows for the substantial volume changes in Si to occur without compromising the protective SEI. Further electrolyte decomposition/electrode degradation is therefore impeded.

This is a careful and interesting study. The use of computational and experimental methods in combination provides a convincing picture of the formation of LiF and organic compounds as a function of voltage and electrolyte composition. The results confirm some previously suspected facts about FEC addition and further clarify the utility of its inclusion. Furthermore the manuscript is well written and engaging. The methods (as pointed out by the authors) are operando XAS study is a tour de force demonstration of the power of this methodology and will surely be adopted by other researchers and used for a variety of electrochemical systems. I highly recommend publication in Nature Communications.

We very much appreciate the reviewer's positive appraisal of our work, and succinctly highlighting the areas where it provides new insights.

I have one tiny comment that the authors may wish to address. In Figure 6, there are two data sets at 1.0V. Like many readers will, I looked at the Figure before reading the text and spent way too much time trying to figure out what the lower (15 minutes later) line was since it is not labeled. After reading the text, it is clear, but not all readers are referees and may be skimming. It would be helpful to label the second curve something like "1.0V after 15 min). Or alternatively, it could be added to the caption. This would allow a more casual reader to understand this plot, which is crucial to the manuscript.

We appreciate the suggestion which is intended to more clearly convey the information within the paper to potential readers. Therefore, we now include a figure label highlighting that the 2nd spectrum at 1.0 V was recorded 15 mins after the previous one. We also include an adjustment to the figure caption to that affect.

Reviewer #3 (Remarks to the Author):

This paper is a very nice piece of Science. It represents a very important advancement in the field of operando spectroscopy, and I do not see any serious reason preventing its publication in Nature Communications. I just have one single question for the Authors. They developed a nice experimental method for separating the current coming from the battery from the photocurrent due to X-ray illumination. Usually this last current is in the range of nanoampere or less, while the current from the battery is many orders of magnitude larger. Could the Authors further comment on this point? I will be quite curious to understand the details of their experiments in this respect.

We appreciate the reviewer's positive assessment of our manuscript, and their confirmation that this is an important advancement in operando methodology. To answer the reviewer's question, typically the current measured from the TEY signal was <5 nA whilst the current under electrochemical cycling was ~250 nA, around two orders of magnitude greater. During voltage holds used for each measurement, the faradaic will vary throughout the measurement. Lock-in based approaches are well suited for extracting small signals from a larger noisy or slowly varying signal and are found to be effective in this application. We now include a statement to this effect when discussing the experimental set-up of the XAS: "A SR830 lock-

in amplifier (Stanford Research Systems) is then used to separate the modulated TEY current (<5 nA) from the faradaic current (~250 nA).”

Reviewer #4 (Remarks to the Author):

Characteristics of SEI layer are always a mystery for the battery community. The lack of understanding of SEI is closely related to the fact of the dynamic nature of SEI layer. Therefore, an ideal way to gain information on SEI is using operando approaches. This manuscript describes operando XAS probing of SEI layer on Si. The key new information appears to be the sequential formation of inorganic (LiF) and organic $-(C=O)O-$ components, which leads to the layer structured SEI. Further, the team also probed the effect of FEC additive, indicating the rapid healing of SEI defects and the improved cycling performance observed. The methodology as described is of general importance for the field of operando study of battery using x-ray absorption. The following points should be considered and clarified.

We thank the reviewer for confirming the general importance of the methodology described and several of the new insights about the SEI that it has been able to provide. We address their more specific points below.

1) Is the Ni thin film pure Ni, or a composite of NiO. If so, how does this affect the deconvolution of O spectra.

The Ni thin film is sputter deposited in a vacuum chamber with Ar as the sputtering gas, leading to the deposition of a metallic Ni film. However, exposure to air on removal from the deposition chamber leads to formation of a surface oxide that is indeed apparent in the O K-edge spectrum as can be seen in figure S5. However, figure S5 also confirms that when 20 nm of a-Si is sputter deposited on top of this layer the feature related to NiO no longer contributes to the O K-edge.

To make this clearer, we have slightly adjusted the text in the SI to more clearly explain why NiO does not contribute to the O K-edge spectra following a-Si deposition:

“This NiO peak is no longer seen when a-Si(20 nm) is deposited on top, consistent with the NiO being buried at a depth greater than the ~10 nm range of electrons detected by TEY-XAS of the O K-edge. It may also be the case that the thin NiO layer is sputtered away to some extent by the energetic Si atoms impinging during sputter-deposition of the a-Si layer.”

We further note that for uncovered Ni films, ongoing TEY-XAS studies for the same electrolyte system have revealed that Ni is fully reduced by 2 V vs. Li. We intend to publish these results in a separate study and believe they fall beyond the scope of this paper. However, this gives confidence that significant NiO contributions are not expected, even for the FY-XAS of Ni films shown in Figure 5.

2) It is apparent that the deposited Si is partially oxidized, therefore, the true structural nature of the Si film needs to be clearly characterized. Is it a Si-SiO_x core-shell morphology, or something else.

The thin film electrodes are sputter deposited as elemental Si, and so are not expected to have a core-shell morphology which would more typically be associated with spherical particles. However, much like the Ni our Si thin films have a native oxide layer due to air exposure following deposition as shown by the TEY-XAS data in Figure S5. The electrochemical cycling displayed in Figure 2 gives confidence that the silicon thin film is amorphous as there are no peaks related to crystalline Si restructuring. Nevertheless, we performed additional Raman spectroscopy of the as-deposited film to further confirm this which is now included in the supporting information (Figure S1, with associated text in the main manuscript: “The amorphous phase of the Si film was confirmed by Raman spectroscopy (see Figure S1), using a Reinshaw inVia Raman microscope with backscattering geometry. A

laser wavelength of 532 nm was used, with a spot diameter of 1-2 μm and an operating power of 0.2 mW focused through an inverted microscope via a 50 \times objective lens.”

3) How does lithiation lead to Si film morphological evolution?

We have now provided SEM data in Fig. 2 (and associated text) to address this point. To briefly summarise, during the first full cycle where the SEI first forms, very little change in electrode morphology is observed. However, more extended cycling (30 cycles) leads to cracking of the Si surface where additional SEI formation is expected. This doesn't affect the conclusions of the paper, where the operando data concerns the first half cycle where significant cracking is not observed.

4) During the lithiation of Si, where is the SEI layer, is it on the film surface or is it penetrated into the film? This essentially relates to question 3).

During the formation of the SEI studied herein, SEM data (now included in Figure 2) indicates the a-Si remains as a continuous film during the first half cycle, and thus the SEI layer forms on the surface of this film which is in contact with the electrolyte. Similar to our response for point 3), we have provided a description of the SEM data that is now included, which addresses this question:

“Figure 2d-g shows SEM of a-Si cells cycled in LP30 electrolyte (no additive) and stopped at different stages of cycling revealing changes in the electrode morphology. Initially, the pristine a-Si electrode (Figure 2d)) appears relatively smooth, showing topography that matches the rolling striations of the underlying Cu substrate. On cycling to 5 mV during the 1st cycle (Figure 2e), the electrode surface remains similarly smooth despite the large expected volume increase due to lithiation of the a-Si, indicating this is primarily accommodated through swelling of the electrode thickness. On cycling back to 2 V (Figure 2f), the overall surface morphology remains largely unchanged although a small amount of cracking can be discerned close to distinct topographic features such as striations. Therefore, during the first cycle the thin film electrode remains continuous with SEI formation expected to occur predominantly at the exposed electrode surface rather than penetrating through the electrode thickness. Close inspection of Figure 2e reveals several small bright dots $\ll 1 \mu\text{m}$ in lateral dimensions, however these are not as apparent in Figure 2f. These are likely products formed during cell disassembly and inert transfer, reflecting the high reactivity of the lithiated a-Si. Following more extended cycling (30 cycles, Figure 2g) much larger morphological changes are apparent with an interconnected network of cracks apparent at 2 V. This is attributable to contraction of the lithiated a-Si as it is delithiated at high potentials leading to the formation of silicon islands of $<1 \mu\text{m}$ in at least one direction, which are separated by sizable gaps of $\sim 200 \text{ nm}$. These provide pathways for electrolyte to penetrate and form fresh SEI through the electrode thickness. These cracks are seen to be refilled to some extent through expansion of the a-Si when it is again lithiated (inset of Figure 2g). This repeated cracking and SEI formation eventually leads to isolation and/or delamination of Si islands from the current collector, contributing to the capacity fade observed after repeated cycling.”

5) It is apparent that the spectra of O is contributed by several sources: Oxygen from SiO_x, oxygen from SEI layer, Oxygen from electrolyte. Therefore, how to distinguish each contribution from the total signal.

As noted in the response to reviewer 1, the overlap of spectral features from different species is a common challenge for experimental spectroscopists, particularly when studying more realistic and complex chemical environments. We have addressed several of the points raised

here in our response to reviewer 1, but provide a further summary here explaining how the different oxygen species are distinguished.

Our assignments of spectral features are based on both reference spectra (both from this work and prior literature), and spectral simulations based on DFT calculations. The spectra of SiO_x and other related references are included in the supplementary information. Simulated spectra of the electrolyte species are presented in the main manuscript and correspond well with prior literature references that are discussed. For the SEI components we have now added additional reference spectra for candidate SEI species. Whilst there are regions of the spectra where large amount of spectral overlap between these different species does occur, we base our analysis on regions where there is minimal spectral overlap and/or where there are characteristic peaks that aren't present in the other species. For example, peak I* is low in energy, and compared to the SiO_x references, the one containing NiO is the only possible overlapping component. However, as explained in our response to point 1, the Ni electrode always becomes reduced upon cycling, thus cannot contribute to this peak at low potentials.

6) The C-edge is not presented, it would be great to show the C-edge information for consistently supporting what is claimed.

We appreciate the reviewer's comment and agree that studies of the C K-edge have the potential to provide further insight into SEI formation in organic electrolytes. However, such studies are made challenging by the prevalence (and accumulation with time) of carbonaceous contamination on many surfaces. This leads to dips in X-ray intensity at the C K-edge for many beamlines due to carbon contamination on optical components such as mirrors. This can completely distort the measured C K-edge signal and can't always be fully removed by I_0 correction. Indeed, for our operando cells it is not immediately obvious how a suitable I_0 measurement could be performed that takes account of adventitious carbon on the outer surface of the silicon nitride windows used to seal them. Additionally, the 100 nm thick silicon nitride windows are much less transparent to incident photons at the energies needed for acquiring the C K-edge compared to the O or F K-edges. Addressing this requires further development of windows which are thinner and/or made from different materials but are still sufficiently strong to maintain the pressure difference between the liquid environment inside the cell and the surrounding vacuum. We hope that this can be achieved in the future, but hopefully this satisfies the reviewer as to why C K-edges are not be included in the present study.

7) Si spectra is very important, while the analysis of Si spectra appears to be rather neglected. One of the key questions is if in addition to the formation of $\text{Li}_{15}\text{Si}_4$, Si appears to be oxidized as well. This aspect of information is missing.

As we noted in our earlier response to reviewer 1, and made clear in the manuscript, the main focus of our study is the SEI formation, rather than the lithiation of Si. We agree however, that there is important information to be obtained from the Si spectra, and we include operando Si K-edge data in the supplementary information (Figure S4) for exactly this reason. We confirm that we have carefully considered and analysed this data, and fully agree with the reviewer that whether the Si remains oxidised at low potentials is a key question. Our data reveals that the Si is reduced by 0.6 V prior to significant lithiation, as seen by the absence of the SiO_2 feature in the Si K-edge compared to the as-deposited a-Si (i.e. air transferred). This data also confirms that lithiation of the Si has occurred by 0.2 V. We directly address this point in the main manuscript:

"Although the focus herein is the SEI components, Si K-edge measurements were also performed at several potentials confirming removal of SiO_2 from the Si surface and the lithiation of Si at low potentials (see supplementary Figure S4), as expected from the cycling data presented in Figure 2."

As pointed out by reviewer 1, we also provide a detailed account of the silicon lithiation based on the electrochemical data of Figure 2, where signatures for the different Si lithiation processes are much more well-established and apparent, in contrast to the SEI formation processes. We thus believe the information requested by the reviewer regarding the Si spectra and chemical state of the Si is included in the manuscript. To address the reviewer's comment we have slightly edited the text in the SI accompanying Figure S4 to highlight the SiO₂ reduction.

Overall, this is a piece of interesting work to the electrochemical community, while associated with the poor spatial resolution of the x-ray based techniques, interpretation of data needs careful deliberation of all possible factors

We thank the reviewer for their positive assessment of our work and the interesting questions they pose which we have addressed above. Whilst we agree that the lateral resolution of our soft XAS approach is limited due to the relatively large X-ray spot size, the interface sensitivity of ~10 nm and ability to resolve different chemical species provide important insights into SEI formation processes. Within this context we have been careful to interpret our data within the limitations of the techniques applied such that robust conclusions are drawn.

Further Comments after Revisions

Reviewer #3 (Remarks to the Author):

The Authors have replied to my previous comment, and I now can recommend this paper for publication.

Reviewer #4 (Remarks to the Author):

Upon revision, the authors have addressed my question very carefully. The revised manuscript is in good standing and I get no objection for publication in the present form.